

# Contrasting growth and mortality responses of different species lead to shifts in tropical montane tree community composition in a warmer climate

Bonaventure Ntirugulirwa[1,2,3,5], Etienne Zibera[4], Nkuba Epaphrodite[3,5] Aloysie Manishimwe[1,2], Donat Nsabimana[4], Johan Uddling[2], Göran Wallin[2,6]

[1]Department of Biology, College of Science and Technology, University of Rwanda, Avenue de l'Armée, Kigali P.O. Box 3900, Rwanda

[2]Department of Biological and Environmental Sciences, University of Gothenburg, P.O. Box 461, SE-405 30 Gothenburg, Sweden

[3]Rwanda Agriculture and Animal Resources Development Board, P.O. Box 5016, Kigali, -Rwanda

[4]School of Forestry and Biodiversity and Biological Sciences, College of Agriculture, Animal Sciences and Veterinary Medicine, University of Rwanda, PO Box 210 Musanze, Rwanda

[5]Rwanda Forestry Authority. P.O.Box 46 Muhanga, Rwanda

[6]Environmental Change Institute, School of Geography and the Environment, University of Oxford, South Parks Road, Oxford, OX1 3QY, United Kingdom

*Correspondence to*: Bonaventure Ntirugulirwa (ntirugulirwabonaventure@gmail.com) and Göran Wallin (goran.wallin@bioenv.gu.se)

**Abstract.** The response of tropical trees and tree communities to climate change is crucial for the carbon storage and biodiversity of the terrestrial biosphere. Trees in tropical montane rainforests (TMFs) are considered particularly vulnerable to climate change, but this hypothesis remains poorly evaluated due to data scarcity. To reduce the knowledge gap on the response of TMFs trees to warming, we established a field experiment along a 1300 - 2400 m elevation gradient in Rwanda. Twenty tree species native to montane forests in East and Central Africa were planted in multispecies plots at three sites along
the gradient. They have overlapping distributions but primarily occur in either transitional rainforest (1600 - 2000 m a.s.l) or mid elevation TMF (2000 - 3000 m a.s.l.), with both early- (ES) and late-successional (LS) species represented in each elevation origin group. Tree growth (diameter and height) and survival were monitored regularly over two years. We found that ES species, especially from lower elevations, grew faster at warmer sites while several of the LS species, especially from higher elevations, did not respond or grew slower. Moreover, a warmer climate increased tree mortality in LS species, but not
much in ES species. ES species with transitional rainforest origin strongly increased in proportion of stand basal area at warmer sites, while TMF species declined, suggesting that lower-elevation ES species will have an advantage over higher-elevation species in a warming climate. The risk of higher-elevation and LS species to become outcompeted by lower-elevation and ES species in a warmer climate has important implications for biodiversity and carbon storage of Afromontane forests.



## 1 Introduction

The global average temperature has already increased by 1.09 °C from the period 1850-1900 to 2020 and is projected to further increase by the end of this century to between 1.4 and 4.4 °C, depending on $CO_2$ emission scenarios, SSP1-1,9 and SSP5-8.5, respectively (Arias et al., 2021). Although warming in the tropical region is predicted to be less than in the boreal region (Arias et al., 2021), the warming sensitivity of tropical trees may be larger since they are adapted to a thermally stable environment (Doughty & Goulden, 2008; Ghalambor et al., 2006; Janzen, 1967) and may have a lower capacity to acclimate to increased

temperature compared to plants from more seasonal regions (Crous et al., 2022; Cunningham & Read, 2002; Way & Oren, 2010).

Forest monitoring data have shown that increasingly warm and dry conditions during recent decades have caused a decline of the Amazon carbon sink (Brienen et al., 2015). More recently, there are indications of this occurring also in the Congo basin

(Hubau et al., 2020). On top of these long-term trends, pronounced negative effects of warm and dry conditions during El Niño years have been observed (Clark et al., 2003; Lewis et al., 2011; Rifai et al., 2018). Both heat and drought contribute to these trends and patterns (Jung et al., 2017), and the two stress factors likely interact such that heat (and the associated increase in water vapor pressure deficit, VPD) increases the severity of drought stress and vice versa (Killi et al., 2017; Zhao et al., 2013; Baumann et al., 2021 & 2022; Humphrey et al, 2021). However, due in part to strong co-variation of heat and drought in the

field, the direct effect of warming on tropical forests remains unclear (Docherty et al 2022; Piao et al., 2020; Wang et al., 2014; Zhou et al., 2013). While negative effects of decreasing precipitation have been demonstrated in stand-scale throughfall exclusion experiments (da Costa et al., 2010; Estiarte et al., 2016; Meir et al., 2014; Nepstad et al., 2007; Rowland et al., 2021), large-scale warming field experiments are lacking. This severely hampers our ability to predict the fate of the tropical carbon sink in a warming world.


Global meta-analyses of experimental data with trees indicate that warming usually has positive effects on leaf net photosynthesis and growth in cooler biomes but negative effects in the tropics (Crous et al., 2022; Lin et al., 2010; Way & Oren, 2010). While warming has an overall negative effect on tropical tree growth across studies (Way & Oren, 2010), it should be emphasized that this is not always the case. Large variability across studies and species has been observed from

positive to negative effects (e.g. Slot & Winter, 2018; Dusenge et al., 2021; Wittemann et al., 2022), possibly due to which tropical temperature zone, e.g. lowland vs highland, the species originate from.

Tropical montane rainforests (TMFs) are considered particularly vulnerable to changes in the environment but our knowledge on how they will respond to climate change is currently very limited (Boehmer, 2011; Salinas et al., 2021). TMFs occur at all

continents within the tropical biome and cover approximately 8% (>1000 m a.s.l.) of the tropical forest area (Spracklen & Righelato, 2014). They provide important ecosystem services, such as regulation of regional climate and hydrology (Alamgir





et al., 2016; Martínez et al., 2009; Soh et al., 2019), storage of surprisingly large amounts of carbon, especially on the African continent (Cuni-Sanchez et al., 2021; Nyirambangutse et al., 2017; Okello et al., 2022), and hosting of high levels of biodiversity and endemism (Kessler & Kluge, 2008). Warming may have smaller direct negative effects on higher-elevation

species growing in cooler TMF environments (at their current distribution) than on lower-elevation species which may already grow near or at their thermal optimum. On the other hand, a warmer climate can make lower-elevation species more competitive at higher elevations, over time shifting species distributions upwards at the expense of current higher-elevation species. Moreover, a warmer climate may shift the competitive balance between species with overlapping distributions, to the disadvantage of those with ranges centred at higher, cooler elevations. Such shift has recently been observed in Andean

montane forests (Duque et al., 2015; Fadrique et al., 2018). The tree community composition shift with warming observed in these studies was caused primarily by increased mortality in higher-elevation species, leading to increasing relative abundance of species with ranges centred at lower elevations, so called "thermophilization". These findings show that there is considerable variation in warming sensitivity among co-existing species, and that this is linked to species distribution range and thermal niche. The underlying mechanisms behind the variation are not well understood, but a recent study on four species from

contrasting elevation origins indicate that species differences in physiological temperature sensitivities play a role (Wittemann et al., 2022).

The recent decline in the Amazon biomass carbon sink was primarily caused by increasing tree mortality and not by decreasing productivity (Brienen et al., 2015). Mortality peaked post-drought, likely as a consequence of weakened maintenance and

defence investments under drought stress (Doughty et al., 2015). As indicated by the Andean studies showing thermophilization (Duque et al., 2015; Fadrique et al., 2018), mortality is likely an important factor affecting tree community composition and biomass also in TMFs, although the mechanisms may be different as the temperatures is lower and precipitation often is higher compared to lowland forests. If traits and growth strategies differ between winners and losers in a new climate, such shifts may have large and long-term impacts on both carbon storage and biodiversity as well as other

ecosystem services (da Costa et al., 2010; Johnson et al., 2016; Uriarte et al., 2016; Zhang et al., 2016).

Tree species dominating in early (ES) and late successional (LS) forests are examples of species groups with different growth strategies (acquisitive vs. conservative, respectively) and possibly different responses to climate change. ES forest has progressively become more common at the expense of LS forest due to land-use changes and now represent >50% of all

tropical forests (Chazdon et al., 2009). However, it remains uncertain if they are winners or loser in a future climate. There are experimental indications that photosynthesis and growth respond more negatively to warming in LS species (Carter et al., 2020; Cheesman & Winter, 2013a; Dusenge et al, 2021; Mujawamariya et al., in review; Slot & Winter, 2018; Tarvianen et al., 2022; Vårhammar et al., 2015). Moreover, field monitoring in a TMF in Kenya showed that the effects of high temperature on tree stem growth was more negative in LS compared to ES and mid-successional species (Gliniars et al., 2013). On the

contrary, ES species, sometimes referred to as fast-growing low wood density species, have been found more suspectable to



climate-induced mortality compared to LS species (Aleixo et al., 2019; Baumann et a1., 2022; Elias et al., 2019; Feng et al., 2017; Herrera-Ramírez et al., 2021). A possible explanation to these discrepancies might be that ES species are more drought sensitive (Elias et al., 2020), while LS species are more heat sensitive.

Knowledge about climate and climate change responses of different tropical tree species is also crucial for tree plantation efforts. Historically, exotic species have been preferred in most tropical areas (Sands, 2005). However, many current plantation programs aim to increase the use of native tree species to diversify and enhance ecosystem services and resilience (Bremer & Farley, 2010; Galhena et al., 2013; Thomas et al., 2014). Plantations with native tree species are generally more similar in habitat structure to natural forests compared to exotic plantations, thereby supporting a more diverse flora and fauna (Stephens

& Wagner, 2007; Brockerhoff et al., 2008). Moreover, multi-species native species plantations may be more resilient to different stress factors (Aguirre-Gutiérrez et al., 2022; Brasier, 2008; Woodcock et al., 2017) and hydrologically preferable for nearby agriculture (Sena et al., 2014) compared to monocultures with fast-growing exotic species. However, the current ambitions to promote the use of native tree species in plantations is hampered by limited knowledge of their climate sensitivity, leading to large risks of failure if species are planted in the wrong place or without consideration of likely climate change

during the coming decades (Aitken et al., 2008; Forrester et al., 2005; Thomas et al., 2014).

Elevation gradients offer the potential to study temperature responses of plants and ecosystems under ecologically realistic conditions (Körner, 2007; Malhi et al., 2010; Tito et al., 2020). However, most studies conducted along tropical elevation gradients have focused on characterizing differences in plant and ecosystem traits of existing natural vegetation, with different

species or ecotypes being present at different elevations (Báez et al., 2015; Cuesta et al., 2017; Fadrique et al., 2018; Malizia et al., 2020; Mujawamariya et al., 2018). While such studies are relevant for exploring how climate shapes plant communities and ecosystems in the long term, they say little about the responses of trees under ongoing rapid climate change. Studies that integrate elevation gradient approaches with experimental manipulations have been suggested to bridge the gap between large-scale observational studies and smaller-scale controlled experiment with limited ecological realism (Sundqvist et al., 2013). A

powerful but underused approach is to use multi-species plantations with identical plant material established at different elevations. If tree species are planted in mixed stands, such studies also offer the possibility to explore climate change effects on tree community composition. That is the approach taken in this study.

We investigated the growth and mortality responses of 20 tree species originating from Afromontane and African highland

forests, planted in large and mixed multi-species plantations along an elevation gradient ranging from 2400 down to 1300 m a.s.l. elevation in Rwanda. The elevation gradient was used as a proxy for possible future warming and irrigation was applied to compensate for variation in rainfall along the gradient. The overall aim of the study was to determine the effect of a warmer climate on tree growth and mortality of tropical montane tree species with contrasting successional strategies and origin





climate. Since this is a stand-level study, we could also assess warming impacts on tree community composition. Our specific
aims were to test the following hypotheses:

H1.   Tree growth responses to a warmer climate are more positive (or less negative) in ES than in LS tree species.
H2.   The warmer climate at lower elevation decreases tree growth in species from higher elevation of origin while it
      stimulates growth of species from lower elevation of origin.
H3.   Mortality increases in a warmer climate, and this is more pronounced in high-elevation and LS species compared to
      lower-elevation and ES tree species.
H4.   Interspecific variation of growth and mortality responses leads to altered tree community composition in a warmer
      climate, favouring lower-elevation and ES tree species.

## 2. Materials and methods

### 2.1 Experimental sites

Growth and mortality of tropical trees were studied within the TRopical Elevation Experiment in Rwanda (Rwanda TREE;
see www.rwandatree.com) during two years after plantation (January 2018 to December 2019). Mixed plantations with 20
tropical tree species native to East & Central Africa were established at three sites along an elevation gradient from 2400 to
1300 m.a.s.l. (Table 1). From the highest to the lowest elevation, over a distance of c. 170 km, the mean annual temperature
(MAT) increased with 5.4 °C while mean annual precipitation (MAP) decreased with approximatley 50%. The high-elevation
site (HE; 2400 m a.s.l.) is located at Sigira in Nyamagabe district (2°30'54" S; 29°23'44" E) in close proximity to the plantation
buffer zone surrounding Nyungwe National Park (NNP), a TMF in the southwestern part of Rwanda. The mid-elevation site
(ME; 1600 m a.s.l.) is located at Rubona agricultural research station in Huye district (2°28'30" S; 29°46'49" E), c. 43 km
south-east from Sigira site. The low-elevation site (LE; 1300 m a.s.l.) is located at Ibanda Makera in Kirehe district (2°6'31"S
;30°51'16" E), c. 126 km north-east of Rubona site, in the eastern part of Rwanda, near the border with Tanzania. The sites
were located in different zones of Potential Natural Vegetation (Kindt et al., 2014): HE in the Afromontane tropical rain forest
or more generaly referred to as the TMF zone; ME in the Lake Victoria transitional rain forests (LVTF) and LE in the evergreen
and semi-evergreen bushland and thicket (Table 1). The Sigira site is considered as the control site in this experiment, as 18
out of 20 species used in this experiment naturally grow in the neighbouring NNP, ranging from 2950 down to 1600 m a.s.l.
(Fisher and Killman, 2008; Nyirambangutse et al., 2017).

### 2.2 Experimental design and plot preparation

Before establishment of plantations, all vegetation consisting of bushes and trees was cleared and large roots were removed.
At each site, 18 plots of 15 m x 15 m, spaced by 2.5 m paths, were established on a 50 x 102.5 m planimetric area. Within each
plot, 20 different species with a replication of 5 (i.e. 100 trees per plot) were planted with 1.5 x 1.5 m spacing and randomized





**Table 1.** Weather and soil characteristics of the sites along the elevation gradient. Weather data show mean ± standard deviation for two years, from 1st February 2018 to 31st January 2020. Soil T and SWC are measured at 10-20 cm depth on 6 plots and other soil data variables represents a mean of 0-30 cm depth of three blocks at each site. Wind, soil T and SWC were only measured during the 2019/20 period. HE, high elevation; ME, mid elevation; LE, Low elevation.

| | Site: | | | P-values |
|---|---|---|---|---|
| | HE (Sigira) | ME (Rubona) | LE (Makera) | |
| *Site characteristics* | | | | |
| Elevation (m a.s.l.) | 2400 | 1600 | 1300 | |
| Latitude | S 2° 30′ 54′′ | S 2° 28′ 30′′ | S 2° 6′ 31′′ | |
| Longitude | E 29° 23′ 44′′ | E 29° 46′ 49′′ | E 30° 51′ 16′′ | |
| PNV | Montane forest | Lake Victoria Transitional rain forest | Evergreen & semi-evergreen bushland and | |
| *Climate* | | | | |
| MAP (mm yr$^{-1}$) | 2144 ± 61 | 1672 ± 136 | 1106 ± 33 | |
| MAT (°C) | 15.2 ± 0.1 | 20.0 ± 0.0 | 20.6 ± 0.1 | |
| T air day mean (°C) | 17.1 ± 0.2 | 22.4 ± 0.1 | 24.0 ± 0.3 | |
| T air night mean (°C) | 13.3 ± 0.1 | 17.5 ± 0.1 | 16.9 ± 0.03 | |
| T air 180 cm 99%ile (°C) | 23.1 ± 0.4 | 28.3 ± 0.3 | 31.3 ± 0.5 | |
| T air 50 cm 99%ile (°C) | 26.1 ± 0.8 | 30.0 ± 0.4 | 32.8 ± 1.1 | |
| T air 180 cm 1%ile (°C) | 10.9 ± 0.3 | 13.7 ± 0.3 | 11.3 ± 0.2 | |
| T air 50 cm 1%ile (°C) | 10.1 ± 0.2 | 13.5 ± 0.2 | 10.7 ± 0.4 | |
| VPD day mean (kPa) | 0.51 ± 0.03 | 1.03 ± 0.01 | 1.14 ± 0.03 | |
| VPD 99%ile (kPa) | 1.45 ± 0.004 | 2.47 ± 0.11 | 2.98 ± 0.16 | |
| PPFD day mean (µmol m$^{-2}$ s$^{-1}$) | 611 ± 66 | 764 ± 62 | 740 ± 31 | |
| PPFD 99%ile (µmol m$^{-2}$ s$^{-1}$) | 1683 ± 134 | 1884 ± 135 | 1841 ± 104 | |
| Wind speed (m s$^{-1}$) | 0.6 | 0.7 | 0.3 | |
| Wind speed gusts (m s$^{-1}$) | 1.2 | 1.2 | 0.7 | |
| Wind speed gusts 99%ile (m s$^{-1}$) | 3.5 | 4.0 | 3.0 | |
| *Soil properties* | | | | |
| Soil T mean (°C) | 16.4 ± 0.6a | 22.0 ± 1.5b | 22.0 ± 1.0b | **<0.001** |
| Soil T 99%ile (°C) | 19.3 ± 1.2a | 27.1 ± 2.5b | 27.5 ± 2.5b | **<0.001** |
| Soil T 1%ile (°C) | 14.6 ± 0.4a | 18.9 ± 0.8b | 18.8 ± 0.9b | **<0.001** |
| SWC (m$^3$ m$^{-3}$) | 0.26 ± 0.07a | 0.19 ± 0.03b | 0.20 ± 0.02ab | **0.024** |
| SWC 99%ile (m$^3$ m$^{-3}$) | 0.36 ± 0.04a | 0.27 ± 0.05b | 0.33 ± 0.03a | **0.005** |
| SWC 1%ile (m$^3$ m$^{-3}$) | 0.18 ± 0.05 | 0.14 ± 0.03 | 0.13 ± 0.02 | 0.068 |
| SBD (g cm$^{-3}$) | 1.0 ± 0.03a | 1.6 ± 0.1b | 1.4 ± 0.1c | **<0.001** |
| Sand (%) | 36.4 ± 2.0a | 56.4 ± 1.4b | 41.4 ± 1.6c | **<0.001** |
| Silt (%) | 19.5 ± 1.7a | 6.5 ± 0.5b | 26.1 ± 0.6c | **<0.001** |
| Clay (%) | 44.1 ± 3.4a | 37.1 ± 1.1b | 32.6 ± 1.1b | **0.002** |
| pH (Water) | 4.2 ± 0.2a | 5.0 ± 0.1b | 6.1 ± 0.1c | **<0.001** |
| pH (KCl) | 3.4 ± 0.1a | 3.9 ± 0.04b | 5.3 ± 0.2c | **<0.001** |
| Al$^{3+}$ (meq/100g) | 4.7 ± 0.4a | 1.4 ± 0.4b | 0.0 ± 0.0c | **<0.001** |
| CEC (meq/100g) | 15.6 ± 2.1a | 8.0 ± 1.0b | 13.9 ± 1.1a | **0.002** |
| Available P (µg g$^{-1}$) | 12.1 ± 3.9 | 14.1 ± 0.7 | 8.2 ± 2.5 | 0.095 |
| N-NH$_4^+$ (µg g$^{-1}$) | 11.3 ± 2.9 | 10.1 ± 2.3 | 7.1 ± 1.0 | 0.133 |
| N-NO$_3^-$ (µg g$^{-1}$) | 28.3 ± 1.9a | 7.2 ± 3.0b | 24.1 ± 4.6a | **0.001** |
| Tot N (mg g$^{-1}$) | 2.73 ± 0.3 a | 2.00 ± 0.4ab | 1.26 ± 0.0b | **0.003** |
| Tot P (mg g$^{-1}$) | 0.65 ± 0.02a | 0.14 ± 0.04b | 0.27 ± 0.02c | **<0.001** |
| Org C (mg g$^{-1}$) | 38.6 ± 1.1a | 27.1 ± 3.1b | 27.3 ± 2.4b | **0.001** |

PNV, Potential Natural Vegetation (Kindt et al., 2014); MAP, mean annual precipitation; MAT, mean annual temperature; T air day, day temperature (PFFD >2 µmol m-2 s-1); T night, night temperature (PFFD <2 µmol m-2 s-1); VPD, Vapor Pressure Deficit; PPFD, Photosynthetic photon flux density. T soil, soil temperature at 10-20 cm depth; SWC, soil water content at 10-20 cm depth; SBD, soil bulk density; CEC, cation exchange capacity; Tot N and P, total nitrogen and phosphorus concertation; Org C, organic carbon concentration.

positioning of each tree individual. The total number of trees was 1800 (i.e. 18 plots x 100 trees) at each site. The 18 plots

allowed for a full factorial experimental design, with three water levels and two fertility levels and a replication of three plots

for each of the six treatment combinations. However, the water and nutrient treatments started in September and November

2019, respectively. No significant (P > 0.05) effect of the treatments was observed during this period likely because September

to December is within the rainy season and we therefore use averages of all plots. Before mid-July 2019, all trees at all sites

received water when needed, irrespective of the subsequent planned water treatment. All trees were manually irrigated

throughout the first dry period in July-August 2018, while all plants were exposed to the dry period from mid-July to end of

August 2019. Site maps with experimental design are presented in Figures S1-S3.

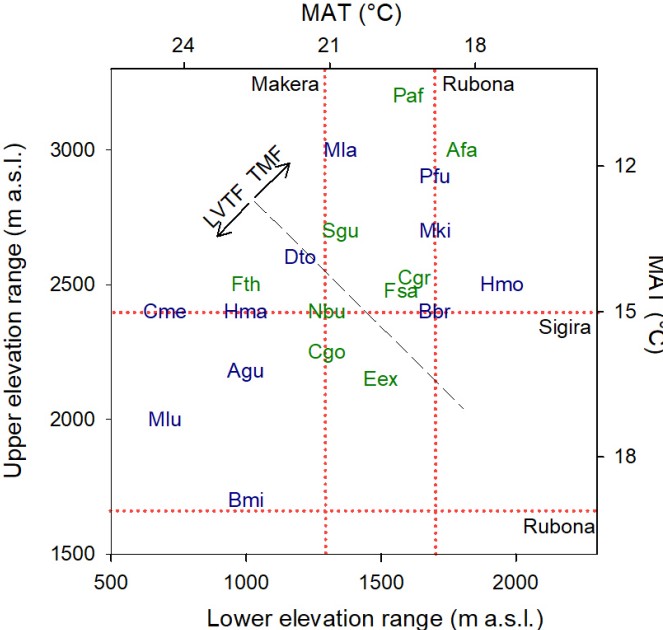

**Figure 1.** Commonly observed natural elevation ranges for the species used in the experiment (Blosch et al., 2009; Fischer &

Killman, 2008; Kindt et al., 2014; Nyirambangutse et al., 2017). The mean annual temperature (MAT) scale was developed

from a linear regression ($R^2 = 0.95$) of MAT versus elevation, using data from eight locations in Rwanda. The black dashed

diagonal line separates the species originating from the tropical montane forest (TMF) and the Lake Victoria transitional forest

(LVTF). The elevation of the experimental sites (high – Sigira; mid – Rubona; low, Makera) are indicated by the red dotted

lines. Species codes in blue and green are early and late successional species, respectively. For full species names, see Table

190 2.





### 2.3 Plant material

The tree species included in the experiment were selected to represent important and common species in two East and Central
African forest types: TMF (> 2000 m a.s.l.) and LVTF (~1600-2000 m a.s.l.), see Figure 1 for species elevation ranges. From
each forest type, species representing both early (ES) and late (LS) succession strategies were selected with the following
number of species per combination: 5 ES/TMF, 5 LS/TMF, 6 ES/LVTF and 4 LS/LVTF (Table 2). Note that from literature,
the classification *Ficus thonninghii* and *Markhamia lutea* into distinct successional groups is ambiguous (Table 2, S4), but
according to our assessment they belong mostly to LS and ES groups, respectively, and will here be treated as such. The
species represent a mix of mostly evergreen and semi-deciduous trees, but also two deciduous species were included (Table
200 2).

The trees were propagated from seeds, cuttings or wildlings, depending on species specific difficulties of propagation, in poly-
pots in a nursery at Rubona research station (near ME-site) during 2017. The germplasms were collected from Nyungwe and
Ndiza TMF (high-elevation) or Rubona research station and Ruhande arboretum located in the LVTF vegetation zone (mid-
elevation) depending on forest type of origin. After six to twelve months in the nursery and having reached a height of ~5-75
cm, depending on species, each plant was randomly assigned to site, plot and plot position (as constrained by the experimental
design) and transplanted to the experimental plots within a period of approximately one month at the turn of the year from
2017 to 2018. Details of the propagation and initial sizes of the trees are given in Tables S1-S3.

### 2.4 Environmental conditions

Weather stations were installed in an open space next to the plantations at all Rwanda TREE sites, equipped with sensors for
measurements of temperature, relative humidity, solar radiation, precipitation and wind direction and speed (VP-3/VP4,
PYR/PAR and ECRN-100, Davis cup anemometer, respectively, from Meter group Inc, Pullman, WA, USA) connected to
data a logger (Em50G, Meter group Inc). In addition, soil temperature and soil water content sensors (Teros 12, Meter group
Inc) were installed in the centre of six plots connected to a data logger (Em60G, Meter group Inc). The recording of most
parameters started late January 2018, while wind and soil measurements started late August 2018 (Table 1). All parameters
were recorded every 30 min. The MAT measured from 1 February 2018 to 31 January 2020 at the HE-site (Sigira), ME-site
(Rubona) and LE-site (Makera) were 15.2, 20.0 and 20.6 °C, respectively. The daytime and extreme temperatures (expressed
as 99%ile) were 17.1/23.1 °C at HE-site, 22.4/28.4 °C at ME-site, and 24.0/31.2 °C at LE-site. The larger difference in the
daytime and extreme temperatures compared to MAT between the ME and LE sites was mainly due to a larger diurnal range
and lower nighttime temperatures at the LE site. Also MAP differed between sites, decreasing progressively from Sigira (2144
mm) to Rubona (1672 mm) and Makera (1106 mm; Table 1). However, the relative seasonal distribution of precipitation was
similar at all sites, with highest rainfall in March–May and a dry period in June–August. Solar radiation was similar at ME and
LE sites while the HE-site received on average less radiation, probably due to higher cloudiness as only small differences in





**Table 2.** Taxonomy of species, their functional type and ecological and geographical distribution. TMF, tropical montane forest (~ > 2000 m a. s. l.); LVTF, Lake Victoria Transitional Forest (~ 1600 - 2000 m a. s. l.), ES, early successional; LS, late successional; The distribution is classified into W, west; C, central; E, east and S, south Africa.

| Code | Scientific name and Author[1] | Family name[1] | Forest type[2] | Succes-sional group[3] | Plant functional type[5] | Distri-bution[6] |
|---|---|---|---|---|---|---|
| Afa | *Afrocarpus falcatus* (Thunb.) C.N.Page | Podocarpaceae | TMF | LS | Evergreen | E |
| Agu | *Albizia gummifera (J.F.Gmel.)* C.A.Sm. | Fabaceae | LVTF | ES | Deciduous | C & E |
| Bbr | *Bridelia brideliifolia* (Pax) Fedde | Phyllanthaceae, (Euphorbiaceae) | TMF | ES | Semi-deciduous | C |
| Bmi | *Bridelia micrantha* (Hochst.) Baill | Phyllanthaceae, (Euphorbiaceae) | LVTF | ES | Semi-deciduous | W & E |
| Cgo | *Chrysophyllum gorungosanum* Engl. | Sapotaceae | LVTF | LS | Evergreen | E |
| Cgr | *Carapa grandiflora* Sprague | Meliaceae | TMF | LS | Evergreen | E |
| Cme | *Croton megalocarpus* Hutch. | Euphorbiaceae | LVTF | ES | Semi-deciduous | C & E |
| Dto | *Dombeya torrida* (J.F.Gmeel.) Bamps | Malvaceae, (Sterculiaceae) | LVTF | ES | Semi-deciduous | W, C & E |
| Eex | *Entandrophragma excelsum* Sprague | Meliaceae | LVTF | LS | Evergreen | C & E |
| Fsa | *Faurea saligna* Harv. | Proteaceae | TMF | LS | Evergreen | E & S |
| Fth | *Ficus thonningii* Blume | Moraceae | LVTF | LS[4] | Deciduous | W, C & E |
| Hma | *Harungana madagascariensis* Lam.ex.Poir | Clusiaceae | LVTF | ES | Semi-deciduous | W, E & S |
| Hmo | *Harungana montana* Spirlet | Hypericaceae | TMF | ES | Semi-deciduous | C |
| Mki | *Macaranga kilimandscharica* Pax | Euphorbiaceae | TMF | ES | Semi-deciduous | C & E |
| Mla | *Maesa lanceolata* Forssk. | Primulaceae | TMF | ES | Semi-deciduous | E |
| Mlu | *Markhamia lutea* K.Schum. | Bignoniaceae | LVTF | ES[4] | Evergreen | E |
| Nbu | *Newtonia buchananii* (Baker f.) G.C.C.Gilbert & Boutique | Fabaceae | LVTF | LS | Evergreen | W, C & E |
| Paf | *Prunus africana* (Hook.f.) Kalkman | Rosaceae | TMF | LS | Evergreen | E |
| Pfu | *Polyscias fulva* (Hiern) Harms | Araliaceae | TMF | ES | Semi-deciduous | E |
| Sgu | *Syzygium guineense* DC. | Myrtaceae | TMF | LS | Evergreen | E |

[1]Taxonomy information from The World Flora Online (http://www.worldfloraonline.org). Family names are given according to Angiosperm Phylogeny Group (APG III) system and when applicable earlier family names is given in brackets; [2]Forest types follows the Potential Natural Vegetation's by Kindt et al. (2014); [3]Sources of successional group classification are given in the Table S4; [4]The classification of Fth and Mlu into distinct successional groups is ambiguous, but according to our assessment they belong mostly to the LS and ES groups, respectively (see table S4), and will be treated as such; [5]Semi-deciduous species drop variable amount of leaf mainly depending drought period severity, but are rarely completely defoliated; [6]According to our definition, central and east Africa distribution areas overlaps in Rwanda, thus trees with both C and E distributions will be found there, but only the main distribution areas are given.

monthly maximum radiation was observed between the sites (not shown). Soil temperatures were closely related to the site MAT, although it was probably also affected by radiation and canopy cover. Soil water content (SWC) was similar at LE and ME sites in spite of differences in MAP and soil texture (Table 1). The mean SWC at HE was substantially higher compared





to the two other sites, probably because of both higher MAP, higher water-holding capacity due to higher soil clay content
(Table 1), and lower evapotranspiration due to lower air temperature.


### 2.5 Soil characteristics

Before planting the trees, in November 2017 soil samples were collected at 0-10 cm and 20-30 cm in the center of each plot
using soil sample rings (Ø 53 mm, Ejkelkamp soil & water, Giesbeek, the Netherlands). Each sample was sealed at site and
later combined for two or six plots in the laboratory, depending on parameter to be analysed. Soil bulk density (SBD), $NH_4^+$,

$NO_3^-$ and available P were analysed from composite samples of two adjacent plots, while the remaining soil parameters were
analysed from composite samples of six plots. This resulted in nine or three composite samples per site, however all values
presented here are averages of 0-30 cm depth and as mean and standard deviation of three samples per site (each combining
six plots), representing three blocks per site (Table 1). Oven dried (70 °C) soil samples of know volume was used to determine
SBD. Samples for analysis of total N and P were pre-grinded using a mortar and pestle and further grinded to a fine powder

with a ball mill (model MM 301, Retsch, Haan, Germany). Soil N and P concentrations were determined by dry combustion
using an elemental analyser (EA 1108, Fison Instruments, Rodano, Italy) and inductively coupled plasma mass spectrometry
(AQ250; ACME Analytical Laboratories, Vancouver, BC, Canada), respectively. For all other parameters, fresh soil samples
were sent for analysis to the Soil & Plant Analytical Laboratory at Rwanda Agriculture and Animal Resource Development
Board (RAB) at Rubona research station, Rwanda.


The main soil texture differences between sites were a larger proportion of sand and less silt at Rubona (53-62% and 5-9%,
respectively) compared to the soil at Sigira and Makera sites (35-45% and 15-27%, respectively) while the clay content was
relatively high at all sites (30-50%). The soil pH (water) was 4.2 at the HE-site and increased with approximately one unit for
each step down the elevation gradient, which explain the decline in Al3+ with decreasing elevation but did not affect the cation

exchange capacity (CEC). Soil fertility expressed as total N and P content tended to decline with decreasing elevation, partly
explained by higher organic content at the HE site compared to the other sites. However, the available N ($NH_4^+$, $NO_3^-$) and P
were not significantly affected by elevation, but $NO_3^-$ was lower at the ME site.

### 2.6 Growth and mortality monitoring

The stem height ($h$) and diameter ($D$) of each tree were measured trimonthly during the first two years, starting at planting

(December 2017/January 2018) and resulting in 8 censuses. The measurement of $D$ was taken using callipers while h was
measured with a measuring stick or, for larger trees, a telescopic measuring pole. For species having several stems, only the
main stem (tallest) was measured regularly and presented in this study, but we also report the average number of living stems
for each species observed at the 7th census (late September to early October 2019). At each census, dead trees or trees with
reduced vitality were recorded. The $D$ measurements of trees with a $h$ below and above 250 cm diameter were done at stem





base (~5 cm above ground, $D_{base}$) and at breast height diameter ($D_{bh}$), respectively. To compare the diameter of all trees irrespective of $h$, parallel measurements of $D_{bh}$ and $D_{base}$ were made at two consecutive censuses after the trees had reached a $h$ of 250 cm as well as after three years. These data were used to develop species-specific functions to estimate the $D_{base}$ for all trees on which $D_{base}$ was not measured and at all census occasions using the following equation:

$$D_{base} = \frac{D_{bh}}{k \cdot h + a} \qquad (1)$$

where k and a are species-specific constants determined from change in $D_{bh}/D_{base}$ ratio with $h$. Note that when roots were causing stem irregularities (swelling etc.) the $D_{base}$ was measured above 5 cm from ground. As a quality assurance, all $D$ and $h$ data were plotted against time to identify outliers and missing values which were replaced by interpolated values. The interpolated values corresponded to <1.5% of the $D_{base}$ and < 0.5% of the $h$ values, out of 43 200 recordings for each variable.

$$RGR = \frac{lnX_2 - lnX_1}{t_2 - t_1} \qquad (2)$$

## 2.7 Relative growth rate of $D_{base}$ and $h$

The relative growth rate (RGR) of both $D_{base}$ and $h$ of the trees were calculated using the standard RGR equation:

where $X_1$ and $X_2$, corresponds to $D_{base}$ or $h$ of the beginning and end, respectively, of each trimonthly census time interval (t1 and t2, respectively). Since RGR varies with tree size (Figure S4) the values were size standardized to be able to make size independent RGR comparisons of different species and sites. This was done for one small and one large tree size category, by averaging the RGR for $D_{base}$ within ranges of 10-25 mm and 50-75 mm (D-RGR$_{D10-25}$ and D-RGR$_{D50-75}$) and for $h$ within ranges of 75-100 cm and 250-300 cm (H-RGR$_{H75-100}$ and H-RGR$_{H250-300}$), respectively.

## 2.8 Community composition

The site effect on community composition was analysed by comparing the basal area (BA) of the four species groups, ES/TMF, ES/LVTF, LS/TMF and LS/LVTF. The BA is a stand level measure defined as the sum of cross-sectional stem area (Ac, calculated from $D_{base}$ values) expressed per ground area. The BA of each species group was calculated as follows:

$$BA = \frac{\sum_{i=1}^{n} A_{ci}}{A_p \frac{n+d}{100}} \qquad (3)$$

where $A_p$ is the plot area (225 m$^2$); $n$ and $d$ are the numbers of living and dead individuals per plot, respectively, within each species group. The term $n+d$/100 is to normalize for initial difference in number of individuals (i.e. number of species x 5 replicates per plot) between species groups (Table 2) but also to include both living and dead trees in the effect on BA. Absolute (cm$^2$ m$^{-2}$) and fractional (% of plot area) BA for each species group are presented as BA$_{abs}$ and BA$_{frac}$, respectively. The influence of mortality on BA for the four species groups was analysed by comparing the result from Eq. 3 with and without the term d. Although, this will not consider the dynamic effect of neighbouring trees exploring the space of the dead trees and

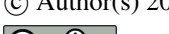



thus potentially having larger $D_{base}$ growth, it will give a reasonable estimate of the importance of mortality for BA of different species groups.

**2.9 Statistics**

The site and species effect on $D_{base}$, $h$, number of stems per individual, tree mortality, D-RGR$_{D10-25}$, D-RGR$_{D10-25}$, H-RGR$_{H75-100}$, H-RGR$_{H250-300}$ were analysed using a two-way ANOVA with site and species/species group as fixed factors. The site and species group effect on BA$_{abs}$ and BA$_{frac}$ were analysed using a two-way split-plot (split between the four species groups within each plot) ANOVA with species groups as within and site as between subject factors. In case of significant site x
species/species group interactions, the site effect was analysed separately for each species/species group, using a one-way ANOVA followed by Tukey's post-hoc test. This latter type of analysis was made also for soil variables. All effects were analysed by using plot level means (n=18 or less when applicable per site). Test of normality (Shapiro-Wilk) was performed to test the distribution of data, and Levene's tested for homogeneity of variance. Normality was mostly obtained for analysis of individual species and sites. Homogeneity of variance were mostly obtained for analysis of individual species and species
groups (for BA) while not when all species or species groups were analysed together. Effects were considered statistically significant at P < 0.05 if homogeneity of variance was obtained and at P < 0.01 when homogeneity was not obtained. Furthermore, to reduce the potential bias caused by non-homogeneity, species with an unequal replication of RGR data on plot level (>1.5 ratio of highest and lowest number of plot replicates between sites) were not included in the statistical analysis. This reduced the number of species included in the standardized RGR analysis and was denoted 'not applicable' (na) in the
presentation of statistical results. Greenhouse-Geisser correction was used to adjust degrees of freedom (df) for within subject factors violating the sphericity assumption according to Mauchly's test. All statistical tests were made by using the SPSS 28.0 software package (SPSS, Inc., Chicago, IL, USA).

**3. Results**

**3.1 Overall effects of high temperature on growth and mortality**

Both site and species identity affected tree growth, expressed as absolute and relative stem $D$ and $h$ increment (Table 3; Figures 2a, b and 3). Tree mortality also differed between sites and species (Figure 2d). However, the responses to a warmer climate (i.e. site) was species specific, as shown by the highly significant (P<0.001) interactions between species and sites for all analysed growth-related variables (Table 3). In the following, we will address how these species differences relate to contrasting successional strategy (ES versus LS) and climate of species origin (TMF versus LVTF).


long




**Table 3.** Results from two-way ANOVA for site and species (Sp.) effects on stem diameter at base ($D_{base}$); stem height; number of stems per individual (stems#), tree mortality, absolute basal area (BA) of species categories as well as on standardised

relative growth rates (RGR) of $D_{base}$ at a $D_{base}$ of 10-25 mm (D-RGR$_{D10-25}$) and 50-75 mm (D-RGR$_{D10-25}$) and of height at a height of 75-100 cm (H-RGR$_{H75-100}$) and 250-300 cm (H-RGR$_{H250-300}$). The analysis was based on plot averages of each species for all variables. Species that did not meet the criterion for balanced number of plot replicates (i.e., at least one individual on each of ≥12 plots per site) were not included in the analysis. Sp#, number of species; n.a., not applicable. See Figure 2 and 3 for post hoc tests.

| Variable | Sp# | Variance Statistics | Source: Site | Sp. | Site * Sp. | Error | Total |
|---|---|---|---|---|---|---|---|
| *Two year after plantation:* | | | | | | | |
| $D_{base}$ | 20 | df | 2 | 19 | 38 | 1017 | 1077 |
| | | F-ratio | 160.4 | 221.9 | 31.4 | | |
| | | P-value | <0.001 | <0.001 | <0.001 | | |
| Height | 20 | df | 2 | 19 | 38 | 1017 | 1077 |
| | | F-ratio | 274.2 | 214.0 | 25.9 | | |
| | | P-value | <0.001 | <0.001 | <0.001 | | |
| Stems# | 20 | df | 2 | 19 | 38 | 1017 | 1077 |
| | | F-ratio | 23.0 | 64.4 | 5.4 | | |
| | | P-value | <0.001 | <0.001 | <0.001 | | |
| Mortality | 20 | df | 2 | 19 | 38 | 1020 | 1080 |
| | | F-ratio | 164.9 | 61.4 | 20.7 | | |
| | | P-value | <0.001 | <0.001 | <0.001 | | |
| *Standardised RGR:* | | | | | | | |
| D-RGR$_{D10-25}$ | 19 | df | 2.0 | 18.0 | 36.0 | 956 | 1013 |
| | | F-ratio | 215.2 | 150.4 | 26.6 | | |
| | | P-value | <0.001 | <0.001 | <0.001 | | |
| D-RGR$_{D75-100}$ | 15 | df | 2.0 | 12.0 | 24.0 | 631 | 670 |
| | | F-ratio | 4.2 | 5.5 | 5.8 | | |
| | | P-value | 0.016 | <0.001 | <0.001 | | |
| H-RGR$_{H75-100}$ | 16 | df | 2.0 | 16.0 | 32.0 | 821 | 872 |
| | | F-ratio | 113.3 | 36.8 | 8.1 | | |
| | | P-value | <0.001 | <0.001 | <0.001 | | |
| H-RGR$_{H250-300}$ | 12 | df | 2.0 | 12.0 | 24.0 | 653 | 692 |
| | | F-ratio | 43.2 | 8.1 | 3.4 | | |
| | | P-value | <0.001 | <0.001 | <0.001 | | |



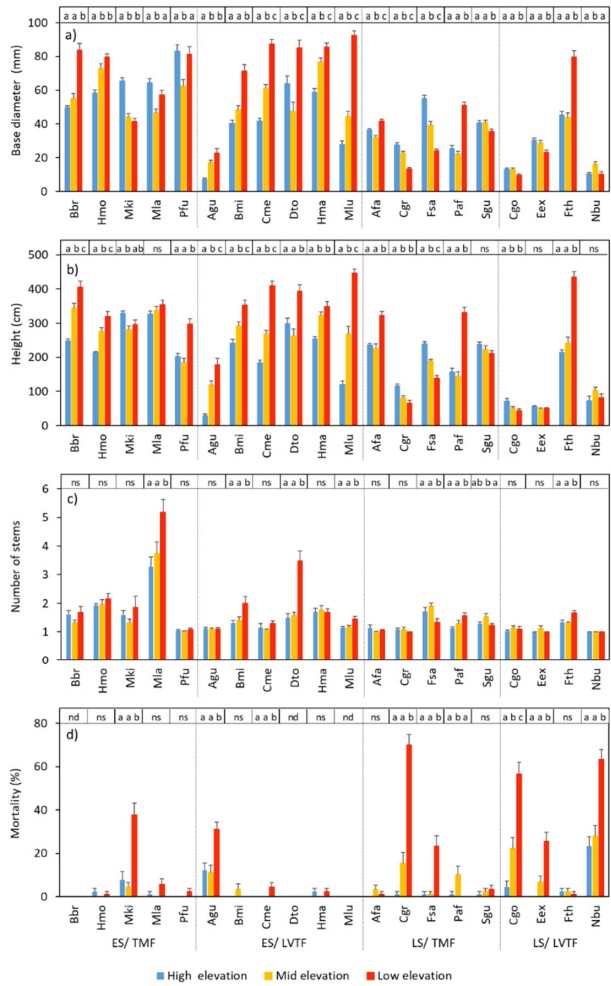

**Figure 2 a-d**. (a) Stem height, (b) stem diameter at base, (c) number of stems from ground per individual and (d) mortality in percent of planted individuals, two years after plantation (except number of stems that was recorded 21 months after plantation) of each species and site (High, Mid and Low elevation). Each bar shows plot level mean with standard error. For tree individuals having several stems, only the tallest stem was measured for diameter and height on each individual tree (panel a and b). Full species names are given in Table 2. Species are grouped into combinations of: ES, early successional; LS, late successional; TMF, tropical montane rain forest; LVTF, Lake Victoria transitional rain forest. The results of post-hoc test (Tukey's) for site effects for each species are reported at the top of each panel. Different letters indicate significant differences (P<0.05); ns, not significant; nd, no dead individuals (panel d) and thus no variance analysis could be conducted. For statistical details, see Table S4.


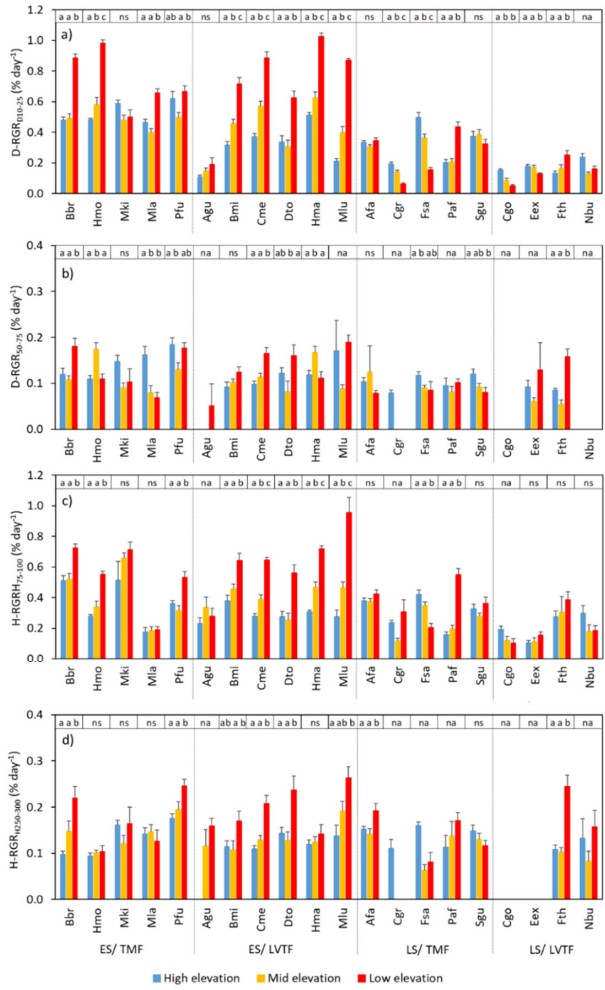

**Figure 3 a-d.** Standardised relative growth rates (RGR) of stem diameter at base ($D_{base}$) at a $D_{base}$ of (a) 10-25 mm (D-RGR$_{D10-25}$) and (b) 50-75 mm (D-RGR$_{D10-25}$) and of height at a height of (c) 75-100 cm (H-RGR$_{H75-100}$) and (d) 250-300 cm (H-RGR$_{H250-300}$) of each species and site (High, Mid and Low elevation). Each bar shows plot level mean with standard error. Full species
names are given in Table 2. Species are grouped into combinations of: ES, early successional; LS, late successional; TMF, tropical montane forest; LVTF, Lake Victoria transitional forest. The results of post-hoc test (Tukey's) for site effects for each species are reported at the top of each panel. Different letters indicate significant differences (P<0.05); ns, not significant; na, not applicable for the statistical analysis, due to too few replicates within the standardised diameter or height intervals. For statistical details, see Table S4 and for standardised RGR details, see Figure S1.






### 3.2 Growth of stem base diameter and height

The most typical growth curve patterns are illustrated for stem $h$ in Figure 4, starting with a lag phase after plantation, followed by faster growth rates that decline with increasing size. As a result of species and site interactions the following growth trajectories were identified: i) increasing growth with warming in both ME and LE sites or only at the LE site, compared to

the HE site (Figures 4a, b), ii) decreasing growth with warming (Figure 4c) and iii) no effect on growth by warming (Figure 4d). For a comprehensive view of growth trajectories of both $D_{base}$ and $h$ of all species, see Figures S5 and S6. For ES species, warmer growth temperature significantly increased $D_{base}$ and $h$ growth in 8 out of 11 (8/11) and 9/11 species, respectively, but the response differed between elevation origin groups (Figures 2a, b; Table S5). A warmer climate significantly increased ES stem $D$ and $h$ in all (6/6) species of LVTF origin, while ES/TMF species exhibited increased, decreased, or unchanged stem $D$

and $h$ with warming. In LS species, stem $D$ and $h$ increased at warmer sites only in 3 out of 9 species, while others significantly decreased or were unaffected. In general, the LS species grew slower compared to the ES species at all sites and the responses to warming in LS species did not depend on species elevation origin. The effects of warm climate on stem $D$ and $h$ were strongly linked, except for *Albizia gummifera* which deviated substantially from the 1:1 line (Figure 5). On average, $D_{base}$ and $h$ after two years for ES species were +12% and +43% at the ME and LE sites, respectively, compared to the HE site, while

corresponding values were -8% and +11% for LS species, -7% and +10% for TMF species and +23% and +59% for LVTF species.

### 3.3 Number of stems

For translation into stem volume production, also the multistem behaviour is important. The individuals of most tree species had one or two stems (Figure 2c). Eight species (5 LS and 3 LS) have, on average, approximately one stem per tree at all sites.

In 6/20 species, trees at warmer sites had significantly higher average number of stems than at the coolest site (Figure 2c). These six species belong to all four groups of successional and elevation origin combinations, but the increase at warm sites was particularly large in the two ES species, *Maesa lanceolata* and *Dombeya torrida*. Noteworthy is that $D_{base}$ and $h$ of M. lanceolate did not respond to warming, instead warming increased wood volume through a significant increase in the number of stems, meaning that standing stock of 10 out of 11 ES species responded positively to warming. Only the LS/TMF species

*Faurea saligna* had a lower average number of stems at the warmest site.


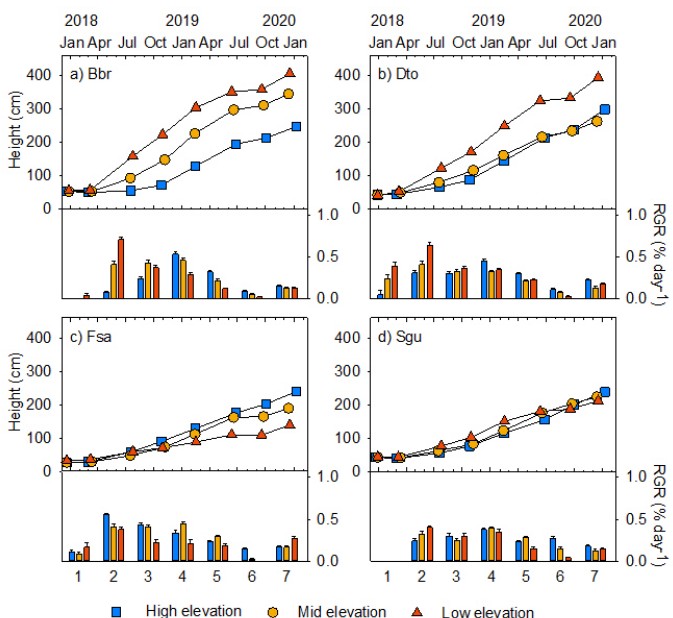

**Figure 4.** Examples of changes in stem height and relative growth rate (RGR) over 2 years, shown for four species at three sites of different elevation: (a) *Bridelia brideliifolia* (Bbr); (b) *Dombeya torrida* (Dto); (c) *Faurea saligna* (Fsa); (d) *Syzygium guineense* (Sgu). See Figures S5 and S6 for growth trajectories of all species.


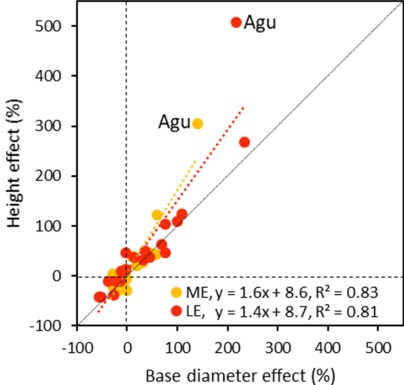

**Figure 5.** The stem height *versus* base diameter at mid (ME) or low elevation (LE) sites compared to the high elevation site, two years after plantation. Each marker is a species mean ($n = 20$). The two values that deviate most from the 1:1 line both belongs to *Albizia gummifera* (Agu).




### 3.4 Standardised RGR of stem diameter and height

Warmer growth temperature significantly stimulated the standardised RGR for smaller trees (10-25 mm $D_{base}$ and 75-100 cm $h$) of both stem $D_{base}$ (9/11) and $h$ (8/11) in ES species, respectively, which is similar to the effect on $D_{base}$ and $h$ after two years (c.f. Figures 2a, b and Figures 3a, c). As expected, the standardised RGR was generally lower for the LS species and very few

LS species grew faster in at warmer sites. Instead, most of them grew slower or were not significantly affected. However, note that 1 and 2 out of 9 LS species did not qualify for statistical analysis of D-RGR and H-RGR, respectively, due to low replication as a result of small sizes and high mortality (Table S5). Taken together, the similarities between warming effects on standing stock after two years and RGR indicate that the different initial sizes of the trees at planting (Tables S2, S3) do not appear to have had a systematic effect on the warming responses.


For larger trees, fewer individuals reached the size ranges (50-75 mm $D_{base}$ and 250-300 cm $h$) which resulted in fewer plot replicates and fewer species that could be included in the analysis than for the smaller trees (Figures 3b, d, Table S5). However, the main picture is that both D-RGR and H-RGR for larger trees were lower than for smaller trees and that fewer large ES species trees were significantly stimulated by warming. This resulted in smaller differences between ES and LS species for

larger trees compared to smaller trees, both regarding overall RGR and the warming responses. This may partly be influenced by a drought period effect in 2019, when no irrigation was applied, causing a particularly negative effect on the larger trees (usually ES) at the warmer sites with higher VPD and lower soil water content (Table 1), combined with possible higher hydraulic vulnerability among ES trees.

### 3.5 Tree mortality

Warmer growth temperature increased tree mortality, especially at the warmest site (Figure 2d). Mortality >10% over the two years period were found in 7 species: 5/9 LS species and 2/11 ES species. High mortality was equally common in species of both TMF (3/10) and LVTF (4/10) origin. The highest mortality was observed in the three LS species *Carapa grandiflora*, *Chrysophyllum gorungosanum*, *Newtonia buchananii* (70%, 62% and 58%, respectively, at the warmest site). Moreover, LS species having high mortality (>20%) usually (4/5) had lower $D_{base}$ and D-RGR at warmer sites (c.f. Figures 2a, 2d and 3a).

The largest part of the mortality at the warmest site (76%) was recorded between July 2018 and June 2019 (Table S6). High mortality may partly be linked to small initial plant size, especially for the two ES species *A. gummifera* and *Macaranga kilimandscharica* which were substantially smaller at planting compared to other species (Tables S2, S3). However, within species, it was predominantly not the small trees that died (data not shown). Furthermore, one of the LS species with the highest warming-induced mortality (C. grandiflora) did not have smaller initial plant size than most other species in the

experiment, so size is likely not the main explanation here. Another potential cause to the high mortality at the LE site is species-specific difficulties for plant establishment in high temperatures. However, it should be noted that only six trees were recorded dead after the first three critical months. Five of these were replaced by new individuals from the nursery and was





not included in the mortality data (Table S6). The fact that species with high mortality also responded negatively to warming in terms of growth rather suggest that species heat sensitivity was the main cause of warming-induced mortality.

**3.6 Community composition**

Tree community composition, expressed both as $BA_{abs}$ and $BA_{frac}$ of the four species groups (ES/TMF; ES/LVTF; LS/TMF; LS/LVTF) and resulting from changes in both $D_{base}$ growth and mortality, was significantly affected by site (Tables 4, 5; Figure 6). The $D_{base}$ growth contributed most to the site mean BA, while mortality affected BA by only 3.4% and 4.5% at HE and ME sites, respectively, but by 17% at the LE site (data not shown). $BA_{abs}$ of LVTF species was significantly stimulated at the LE
compared to the HE site, while TMF species were unaffected or showed decreased values (Figure 6a). The $BA_{abs}$ of the different species groups at the ME site were similar to either the LE (LS/TMF) or the HE sites (the other three groups). For $BA_{frac}$, values increased at warmer sites in the ES/LVTF and LS/LVTF groups while it decreased for ES/TMF and LS/TMF. All sites were significantly different from each other within each species groups, except for HE and ME sites in the LS/LVTF group (Figure 6b). With these contrasting responses of different species groups, there was a marked difference in community
composition between the sites (Figure 7). Compared to the coolest site, stands in a warmer climate had smaller BA fractions of TMF species and larger fractions of LVTF species. The differences were particularly large for the ES/LVTF and the LS/TMF groups where the BA fractions were almost doubled or halved, respectively.

**Table 4.** Two-way split plot ANOVA for site and species (Sp.) group effects on absolute (abs) and fractional (frac) basal area
(BA) of four species groups (ES/TMF; ES/LVTF; LS/TMF; LS/LVTF). Greenhouse-Geisser correction was used to calculate the degrees of freedom (df) for within subjects as assumption of sphericity was violated. Note that, between subject analysis was not applicable for $BA_{frac}$ as the sum value is 100% for each plot.

| Variable | Source | df | F-value | P-value |
|---|---|---|---|---|
| $BA_{abs}$ | *Between subject:* | | | |
| | Site | 2 | 869.9 | <0.001 |
| | Error | 51 | 27.7 | <0.001 |
| | *Within subject:* | | | |
| | Sp. group | 1.9 | 372.1 | <0.001 |
| | Sp. group * site | 3.8 | 43.7 | <0.001 |
| | Error | 98 | | |
| $BA_{frac}$ | *Within subject:* | | | |
| | Sp. group | 2.2 | 764.5 | <0.001 |
| | Sp. group * site | 4.4 | 58.0 | <0.001 |
| | Error | 112 | | |





**Table 5**. One-way ANOVA for site effects of absolute (abs) and fractional (frac) basal area (BA) of four species groups (ES/TMF; ES/LVTF; LS/TMF; LS/LVTF). Degrees of freedom are 2, 51 and 54 for site, error and total, respectively. See Figure 6 for post hoc tests.

| Variable | Species group | Site effect: F-ratio | Site effect: P-value |
|---|---|---|---|
| BA$_{abs}$ | ES/TMF | 8.9 | <0.001 |
| | ES/LVTF | 57.7 | <0.001 |
| | LS/TMF | 11.5 | <0.001 |
| | LS/LVTF | 29.3 | <0.001 |
| BA$_{frac}$ | ES/TMF | 46.5 | <0.001 |
| | ES/LVTF | 89.6 | <0.001 |
| | LS/TMF | 67.7 | <0.001 |
| | LS/LVTF | 5.4 | <0.001 |

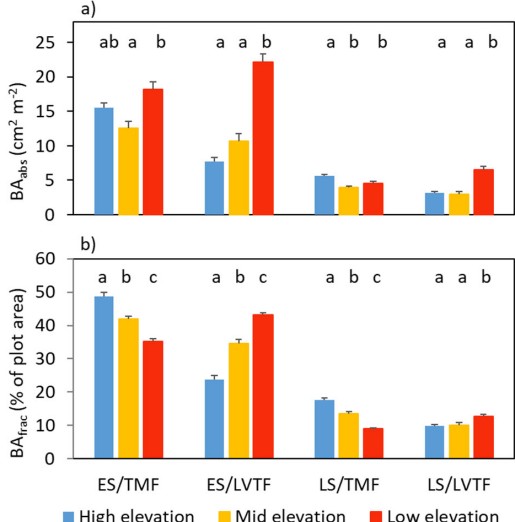

**Figure 6 a, b.** (a) Absolut and (b) fractional basal area (BA$_{abs}$ and BA$_{frac}$, respectivley) after two years per species category of early (ES) and late (LS) successional species from tropical montane forest (TMF) and lake Victoria transitional forest (LVTF) from three sites at different elevations. Both BA$_{abs}$ and BA$_{frac}$ was standardised to compensating for the somewhat different initial number of individuals between the species groups (see eq. 3). Each bar show plot level mean (n=18) with standard error. Different letters above each species category indicate significant difference (P<0.05) between sites.


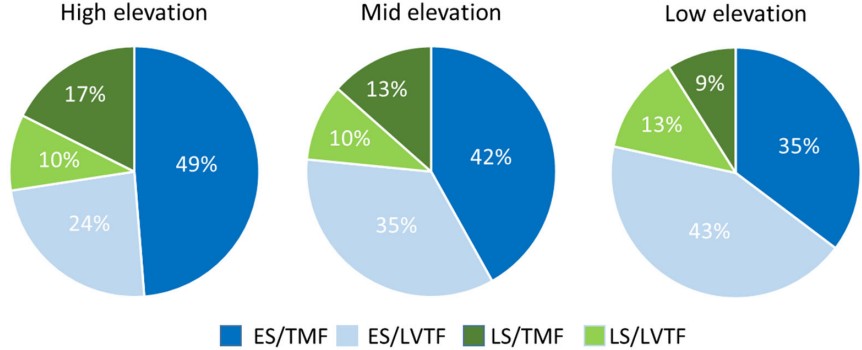

**Figure 7**. Fractions of basal area (BA$_\text{frac}$) two years after planting for four combinations of early (ES) and late (LS) successional species from tropical montane forest (TMF) and lake Victoria transitional forest (LVTF) grown at three sites at different elevations. BA$_\text{frac}$ was standardised to compensating for the somewhat different initial number of individuals between the species groups (see eq. 3).

## 4. Discussion

We used an experimental elevation gradient approach to explore the temperature responses of the growth and mortality of a broad range of tropical tree species with contrasting successional strategies and elevation ranges of origin in the Rwanda TREE project. Current knowledge on the temperature sensitivity of tropical trees is based primarily on either monitoring in intact forests (e.g. Bennett et al., 2021; Clark et al., 2003 & 2010; Damptey et al., 2021; Herrera-Ramírez et al., 2021; Hubau et al., 2020; McDowell et al., 2018; Slot et al., 2021; Sullivan et al., 2020) or controlled warming experiments on small plants grown in chambers (e.g. Cheesman & Winter, 2013a, Cheesman & Winter, 2013b; Slot & Winter, 2018; Wittemann et al. 2022). The semi-controlled experimental approach of Rwanda TREE bridges the gap between these two types of studies by using large plantations with the same species and genotypes grown at three sites spanning 5.4 °C in MAT, with irrigation to compensate for site differences in precipitation. Moreover, by using mixed multi-species tree communities, we could assess the implications of interspecific variation in temperature responses for tree community composition in a warmer climate. This has, to our knowledge, never been explored in an experimental and tropical setting before.

### 4.1 Warming effects on early and late successional species

Warmer climate stimulated the standing stock of stem volume related variables ($D_\text{base}$, $h$ and number of stems) after two years in 10 out of 11 ES species while responses of LS species were mixed, including increases, decreases and no significant changes (Figure 2). Similar pattern were observed for standardized RGR of both $D_\text{base}$ and $h$ (Figure 3, Table S5). These results support





our first hypothesis (H1), stating that the growth of ES species respond more positively to warming than LS tree species. They are in accordance with results from a study with potted seedlings grown in the same soil at the three Rwanda TREE sites,

where warming strongly stimulated biomass growth in the ES species *Harungana montana* but not in the LS species *Syzygium guineense* (Dusenge et al., 2021). This support that the site effects and successional group difference observed here were caused by climate responses rather than by site differences in soil conditions (Table 1). Moreover, Rwandan studies on photosynthesis showed stronger negative effects of high temperature and warming in LS compared to ES tree species (Vårhammar et al., 2015; Tarvainen et al., 2022; Mujawamariya et al., in review). The finding of contrasting warming

responses of ES and LS species is also in line with results on photosynthesis and growth in warming experiments with tropical tree seedlings conducted in indoor climate-controlled chambers (Slot & Winter, 2018) as well as in field chambers (Cheesman & Winter 2013b). Finally, our results are consistent with field observations of negative effects of temperature on stem growth being stronger in LS compared to ES and mid-successional species in a TMF in Kenya (Gliniar et al., 2013). Several lines of evidence thus indicate a shift in competitive balance favouring ES over LS species in a warmer climate.


The loss of competitiveness of LS compared to ES species at warmer sites may in part be linked to the high sun exposure of plants in this study. When plants were small, especially during the first year of the experiment, they were growing in open conditions. This may have been stressful for LS species due to their generally lower tolerance to high light intensities, as observed in previous studies (Poorter, 1999; Veenendaal et al., 1996). However, it is likely not high solar radiation per se that

is harmful for many LS species at the warmer sites in our experiment since peak and mean solar radiation was actually slightly lower at LE compared to the ME site, while growth of several species was significantly lower at the LE compared to the ME site. An indirect effect of heat is more likely, such that the lower inherent stomatal conductance of LS compared to ES species (especially at the LE site; Mujawamariya et al., in review) leads to higher leaf temperatures and stronger physiological heat stress under warm conditions. This has indeed been observed in previous studies in Rwanda (Vårhammar et al., 2015;

Ntawuhiganayo et al., 2020; Tarvainen et al., 2022; Mujawamariya et al., in review). The negative effect of a warmer climate observed in some LS species in this experiment may therefore be stronger than effects that would have occurred in understorey conditions. However, while LS species as young trees often grow in the understorey, they also frequently occur in forest gaps in TMF forests (data from Nyirambangutse, et al. 2017). The lower competitiveness of sun exposed LS seedlings in a warmer climate thus implies an amplified disadvantage at early stand developmental stages compared to ES species. Since early stand

development is a crucial phase for tree community composition, this handicap of young LS seedlings is likely to endure into later stages.

### 4.2 Warming effects on species of different elevation origins

The finding that tree growth of many species is stimulated by warmer climate is contrary to field observations that warm conditions decrease tropical tree growth (Clark et al., 2003, 2010 & 2013; Dong et al., 2012; Feeley et al., 2007; Way & Oren,

2010; Vlam et al., 2014; Hubau et al., 2020). However, this study is on montane and highland tree species that naturally grow



in rather cool conditions, with prevailing mean annual temperatures at the middle of the species elevation ranges of approximatley 15 to 21°C (Figure 1). It is therefore not surprising that tree growth in many species was stimulated at the warmer sites compared to the coolest site, provided that we irrigated during the first 1.5 years to avoid drought stress.

The stimulation of growth rate at warmer sites was especially high for the group of ES species with LVTF origin (+39% and
+140% at the ME and LE sites, respectively, compared to the HE site, Figure 3a). Thus, at least for ES species, results supported the latter part of H2, stating that warmer climate stimulates growth of species from lower elevation of origin. There was also a partial support for the first part of the H2, stating that warming decreased tree growth in species from higher elevation of origin (TMF) as warming had negative effects on tree growth in 5 out of 10 TMF species. Our results are in line with findings in a controlled chamber warming experiment with seedlings of four of the species used in this study, where the lowest-elevation
species was better at physiologically acclimating and growing at high growth temperatures than the highest-elevation species (Wittemann et al., 2022). The differences in warming response of species with different elevation origins (LVFT vs TMF) also corroborates the observation of thermophilization in Andean tropical forests, where tree species with their distribution ranges centred at lower elevation had gained in relative abundance as climate got warmer over time, at the expense of higher-elevation species (Duque et al. 2015; Fadrique et al. 2018). The present study and the chamber warming study on Rwandan tree species
(Wittemann et al., 2022) suggest that warming-induced declines in abundance of higher-elevation species is underpinned by lower capacity of leaf physiology and tree growth processes to handle, or benefit from, warming.

**4.3 Warming effect on tree mortality**

Warmer climate increased tree mortality in LS species, but not often or much in ES species (Figure 2d). However, there was no clear difference between TMF and LVTF species (Figure 2d) and, thus, H3 was therefore only partly supported. The lack
of expected difference between TMF and LVTF conflicts with the observations that thermophilization is driven by higher mortality of species with higher elevation ranges (Duque et al. 2015). It may be that the limited number of species in this study has masked possible influences of species origin, or that such will show up at later stages in the experiment.

The findings of highest mortality in especially the slow-growing LS species (cf. Figure 2a, b & d) are in line with Laurance et
al. (2004a) who observed increased mortality among slow- compared to fast-growing tree species, in an analysis of growth measurements made in 18 one- hectare plots over up to three decades in the Amazon rainforest. However, opposite to these results, more recent long-term studies of mortality in 24 Australian plots and 189 Amazonian plots with tropical forest concluded that fast growing species were at highest risk under increasing drought and high VPD, respectively (Baumann et al 2022; Esquivel-Muelbert et al., 2020). Moreover, studies in Amazonia and Borneo rainforests observed especially high
mortality among ES tree species during hot and dry conditions in El Niño years (Alexio et al., 2019; Slik, 2004). Interestingly, the ES species with high mortality in Slik (2004) belonged to the genus *Macaranga* (seven species) which was the same genus as one of the two ES species with high mortality in our experiment (Figure 2d).



The contrasting results of our study and some of the previous observational studies may be a consequence of ES species being more sensitive to drought while LS species are more sensitive to high temperature and VPD, particularly at young ages. Indeed,

most of the mortality in the observational studies, especially of genera affiliated to wetter climatic regimes, were attributed to drought/water stress (Esquivel-Muelbert et al., 2017). If considering only the effect of warming when there is no water limitation, as in our irrigated study, then ES species might be protected against heat stress by transpiratory cooling while LS species (with lower transpiration) experience stronger physiological heat stress. This was indicated in a previous Rwandan common garden study with seedlings (Vårhammar et al., 2015) as well as in a recent Rwanda TREE study exploring upper

heat tolerance in tropical tree species (Tarvainen et al., 2022). However, the reliance on transpiratory cooling to prevent overheating in ES species may not be a feasible strategy during drought, possibly contributing to the above-mentioned reports of high mortality in ES species.

The higher mortality in ES species in monitoring studies may be likely linked to higher hydraulic vulnerability compared to

LS species (Apgaua et al., 2015; Eller et al., 2018), as well as to their demography with higher recruitment rate and shorter life span (Laurance et al., 2014b; Nyirambangutse et al., 2017). These effects may be mainly expressed by taller mature trees, which may contribute in explaining the difference between our results and long-term observational studies (Baumann et al 2022; Esquivel-Muelbert et al., 2020) Thus, we cannot exclude that the mortality pattern among successional groups may change with increasing age and size of the trees. If LS trees mainly respond negatively to warming in young ages, the

recruitment rate may still be inhibited causing at least a transitional shift in species composition. Consequently, the succession from secondary to primary forest will be slower, with implication for both large-scale carbon storage (across stands at different successional stages) and how fast the final goals of forest restoration can be achieved. Recovery of species composition is already a slow successional process (>century), which may thus be even slower in a warmer climate (Poorter et al., 2021).

**4.4 Community composition**

We showed that ES species with transitional rainforest origin strongly increased their total and fractional basal area at warmer sites, while TMF species declined (Figure 7), supporting H4. These results are in line with the thermophilization observations in Andean tropical forests along elevation gradients, where tree communities have experienced directional shifts in composition towards greater relative abundances of species with ranges centred at lower, warmer elevations during recent decades (Duque et al. 2015; Fadrique et al. 2018). This neotropical thermophilization was caused primarily by increased

mortality of more cool-adapted and heat-sensitive species (with higher elevation ranges), while the community composition shift at warmer sites observed in this study was mostly explained by positive growth responses of species with lower elevation ranges. Contributions by mortality were small since it was highest in species with small trees that do not contribute much to basal area. However, since LS trees grow slower but in the long term grow taller (Laurance et al., 2014b; Nyirambangutse et al., 2017), similar species-specific mortality rates in a mature forest would have had a much larger influence on BA. At later

stand development, it is likely that the increased competitiveness and growth of lower-elevation origin species in a warmer

Biogeosciences Open Access
Discussions
EGU

climate will lead to increased warming-induced mortality of higher-elevation species also in the Rwanda TREE plantations, when competition for resources increase (Lohbeck et al., 2013). It is also possible that the observed growth reduction in higher-elevation LS species at warmer sites is an indication of coming mortality (Cailleret et al., 2016; Esquivel-Muelbert et al., 2020). However, there are several climate related pathways towards mortality (Gora & Esquivel-Muelbert, 2021; Zuleta et al., 2022)

and future research should try to disentangle the mortality mechanisms of different climate variables and how it interacts with different groups of species.

## 5. Conclusion

Like in studies of leaf morphology (Mainshimwe et al., 2022) and photosynthesis (Dusenge et al. 2021; Tarvainen et al., 2022; Wittemann et al. 2022; Mujawamariya et al., in review) in the Rwanda TREE project, the tree growth response to warming

was highly species specific. This points out the importance of including several species representing different ecological niches when assessing warming responses of highly divers tropical forests. We showed that a warmer climate increased tree growth in most ES species while responses of LS species were more variable. Growth stimulation in ES species was stronger in species originating from transitional rainforest compared to higher-elevation TMF. Several of the LS species had both decreased growth and increased mortality at warmer sites. At the stand level, the contribution of TMF species to stand basal area

decreased at lower elevation and warmer climate. We conclude that many tropical highland and montane species show potential growth stimulation in a warmer climate, while other species are likely to be threatened by projected increases in temperature. In particular, higher-elevation and LS species risk to become outcompeted by lower-elevation and ES species in a warmer climate, with important implications for biodiversity and carbon storage of Afromontane forests, as well as for selection of species for re- and afforestation.

**Supplement.** Additional supporting information is found online in the Supplement.

**Data availability statement.** The data that support the findings of this study are available from the corresponding author upon reasonable request.

**Author contributions.** BN, GW, JU and DN planned and designed the experiment; BN, EZ, NE and AM conducted field

measurements; EZ, BN and GW complied and analysed the data; BN, GW and JU wrote the manuscript with contribution from the other authors. All authors read and approved the final manuscript.

**Conflicts of Interest.** The authors declare no conflict of interest.

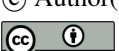



**Funding.** This study was funded by the Swedish research council (VR; grant 2015-03338, 2018-04669 and 2021-05265), Swedish Research Council for Environment, Agricultural Sciences and Spatial Planning (Formas; grant 2015-1458, 2020-01497); JU and GW were also supported by the Swedish strategic research area 'Biodiversity and Ecosystem services in a Changing Climate' (BECC; http://www.becc.lu.se/).

**Acknowledgement.** We are grateful to the Rwanda Agriculture and Animal Resources Development Board (RAB) for providing land for the field plots, and to the Rwanda Development Board (RDB) for providing access permits to seed collection in Nyungwe National Park. Furthermore, we are grateful for all the support from the driver and the field staff at the Rwanda TREE sites.

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
