# Peer review of "Thermophilization of Afromontane forest stands demonstrated in an elevation gradient experiment"

_Biogeosciences, 2023_

## Author Comment (AC1)

**Comments from referee 1 (in blue)**

In this manuscript the authors describe the results of a large manipulative experiment where multiple tree species were planted at three different elevations in order to observe the effect of temperature change in their growth and mortality. In addition, these species represented different successional strategies (Early and Late) and different forests of origin (montane vs transitional). I find the experiment very impressive. The manuscript is very well-written and presented so I want to congratulate the authors. I do however, think that the story is quite complicated and the constant use of acronyms does not help to simplify it. The intro needs a few adjustments to make it shorter and more concise (see below) but it is on the Methods/Results and Discussion where it gets harder to follow and stay engaged. These are a few suggestions to make the manuscript more engaging:

o        Try to reduce the number of acronyms or modify the current ones for something more explicit, for example: LVTF- Transitional forests, TMF – Montane forests; ES- Early-S, LS- Late-S

o        Include a diagram that explains the experimental set up with the three elevations, species origins, data recorded (could be in Supplement)

o        Include a summary figure with your main results. The figures (for example fig.2 and 3) have a lot of information and it is hard to focus on what is important and what does it mean.

Something that I do not understand is the role of the higher elevation site as the control site. That is mentioned in the methods but I do not see much discussion around this fact.

The title undersells the study. That could be the title for an observational study. I think it should reflect the enormous experimental work and novelty of it.

**Response:** Thank you very much for appreciating our study. Thanks also for all constructive suggestions for improvements.

We will try to reduce the use of acronyms in the revised manuscript. We will also try to construct a diagram explaining the experimental set-up for the supplement or as a graphical abstract. We agree that Fig. 2 and 3 have a lot of information, but we have tried to synthesis this information in Fig. 5, 6 and 7 so we believe that additional figures are not needed. The reference to the highest site as the control site is accurate for the species originating from montane rainforest; for these species the other sites represent a warming scenario. For the species originating from transitional rain forest it is not that simple. We will revise to more clearly acknowledge and discuss this matter. Thanks for suggesting a stronger title. As a revised title we suggest: Thermophilization of Afromontane forests demonstrated in an elevation gradient experiment.

Some line comments:

Abstract

Very well written. The only doubt I have is at what stage where the plants transplanted (seedling, sapling…)

**Response:** It was seedlings (except for one species for which cuttings were used but in a "seedling size"). We will add this information in the abstract.

Introduction

45- negative effects on xxxx and where?

**Response:** Thanks for the comments, we will add this information: "*…warm and dry conditions during El Niño years have caused negative effects on tree growth in Central America, Amazonia, West and Central Africa and South-West Asia as well as increased tree mortality in Amazonia (Clark et al., 2003; Lewis et al., 2011; Rifai et al., 2018).*"

50 - in the field? Need to be more specific, In mountains heat and drought do not always covary

**Response:** We will remove "*in the field*" as it may confuse, and more clearly point out that heat and drought not always covary by changing this sentence: "*However, due in part to common co-variation of heat and drought, the direct effect of warming on tropical forests remains unclear*"

59 – Large variability… This sentence is unclear needs to be reorganized.

**Response:** We will revise the sentences as follow: "*Large variability across studies and species has been observed, from positive to negative effects (e.g. Slot & Winter, 2018; Dusenge et al., 2021; Wittemann et al., 2022). This may reflect large variation in species origin temperature zone and how this compares to experimentally applied temperature treatments, with positive effects being more likely if origin climate is rather cool and the warming treatment is modest.*"

64 – You may want to read Tovar et al 2022

**Response:** Thank you for bringing this recent publication to our attention. We will cite this to support the statement on line 64.

65 – The lower elevation limit of TMF varies widely between and within continents so I am not sure what this >1000 m a.s.l. refers to

**Response:** We agree that the TMF elevation limit is highly variable. However, most estimates of their area globally are based on a general lower elevation limit. We think it is important to give the reader an idea of the extension of such forests so we used and relatively recent estimate from Spracklen & Righelato, 2014 who used 1000 m a.s.l. as a limit. Here is preliminary modification of the text to clarify this: "*TMFs occur at all continents within the tropical biome. The lower elevation limit varies widely between and within continents, however, by using a general lower limit of 1000 m a.s.l., the cover has been estimated to 8% of the total tropical forest area (Spracklen & Righelato, 2014).*"

70 – why species on the higher elevations are further away from their thermal optimum? You imply that they could tolerate a higher increase in temperature than lower elevation plants, but you need to justify that. For example, Leon-Garcia and Lasso 2019. Although here they go all the way to the paramo ecosystem.

**Response:** We agree that this statement should be justified by a citation. Thanks for the suggested reference, but we think that Feeley et al 2020 in *Frontiers in Forests and Global Change,* better support this statement so we will add that reference.

83-90 The intro is great until this paragraph. I think it is a bit repetitive, and there is one minor point that could be made more clearly and briefly. Maybe as a first line of the next paragraph a sentence linking the functional strategy with different chances to survive and then growth forms?

**Response:** We agree that this paragraph is a bit redundant and suggest to removing it completely.

o        I am more familiar with using ES and LS to refer to species but primary or old-growth and secondary forests to the forests. I think better to keep them separate.

**Response:** We agree that primary and secondary forests are more commonly used and we will change to this terminology in the revised version.

o        I don't like the sentence: remains uncertain. is it the forests? The species?.

**Response:** We agree that this is unclear and we will change this sentence to: "*However, it remains uncertain if the ES species are winners or loser in a future climate, and thus if the expansion of secondary forests will be amplified or not.*"

o        Also, these experimental indications – how are they different from what you are doing here? Are they greenhouse experiments, models? Better to indicate so you can highlight the novelty of your paper.

**Response:** Thanks for this suggestion. We will to add a sentence that these studies either only included few species representing ES and LS strategies, use artificial heating in chambers or infra-red heaters or only have looked at physiology

100 – defining ES species here is a bit late, should come at the beginning of the paragraph.

**Response:** We will move the text about fast and slow growing species to the beginning where ES and LS species are defined as acquisitive vs. conservative species.

100 – suspectable= susceptible? Susceptible

**Response:** Thank you for spotting this typo. It should be susceptible.

Overall, this paragraph also needs some reorganizing and trimming

**Response:** We think that the paragraph is mostly good, but as said above we will move the information about slow and fast growing species to the first part.

105-115. This seems like a good point but is it important enough to be a whole paragraph? I think the intro is really long and this is something that you should cut.

**Response:** We think it is an important point, but we will try to make it shorter in the revised version

116 – I feel now we are back on track. This brings the story back to line 81 – I would probably talk about the growth strategies after this paragraph or even after the next one where you explain your objective. The reader needs to know early on the intro what you are doing on the manuscript, and it is not clear until here.

**Response:** Thanks for the suggestion, we will consider moving the growth strategies to after this section.

125 – identical plant material- not sure what it means

**Response:** We will change this to "*genetically similar plant material*"

129 – at what stage are the trees transplanted? Saplings?

**Response:** The plants were in seedling sizes when transplanted from a central nursery (at the mid elevation) to the sites but most species developed quickly into sapling sizes at the sites, while a few stayed in a seedling size for quite long time. We prefer not to explain this in the introduction as it

may complicate the overall message of the design. Instead, it is well described in the material and methods as well as in the Table S1 of the supplement.

135 – are lower elevation species planted on even lower elevations? I get that from the hypothesis. Maybe include the elevational range of the species in the objective?

**Response**: The lowest site is below the "transitional rainforest" category so we would say YES, there is a warming treatment for all species. For some species, the distribution range covers the lowest site, but dominating distribution is from the "transitional rainforest" area. We will revise to make this clear already at the end of the introduction.

Overall, I would reduce the background information on precipitation and drought because now I realize that you are irrigating to isolate the effect of temperature, so better to focus on that.

**Response:** We believe that it is important to give background information about both drought and temperature as they are closely connected. In our case the soil drought effect is small (expect the 2019 drought period) while still the there is an VPD effect. I don't understand this comment as it seems related to the introduction.

Methods

150 – increased/decreased by

**Response:** This will be changed accordingly

159 – I don't understand why that site is the control at 2400 because the species grow nearby at (potentially) 1600m which is as low as the mid-elevation site.

**Response:** As pointed out earlier, we will remove this statement to avoid any confusion.

In Table 1 PNV of HE should be the full name to make it easier to link with the text (TMF)

**Response:** This will be changed accordingly

224 – I do wonder about the effect of solar radiation. The HE site is thus the control but also the more shaded one.

**Response:** Yes, solar radiation is slightly lower at the HE site, This is clearly mention in material and methods and possible effects of radiation is discussed on line 491-496.

It is overall a bit confusing the species origin vs the planting sites. And where is the control evaluated?

**Response:** As indicated above, we will revise to more clearly acknowledge and discuss this matter. As mentioned earlier we will remove the statement about control as the results not has been specifically analysed in relation to a control.

Results (see comments above)

Discussion

405 – This drought period, was it a problem with the irrigation system? Has it been mentioned before? can it have an effect on other variables?

**Response:** On line 179-181 it is mentioned: "while all plants were exposed to the dry period from mid-July to end of August 2019." The drought in 2019 was therefore intentional and not linked to a

problem with the irrigation system. As mentioned on line 221-222, the relative seasonal distribution of precipitation was similar at all sites, and the 2019 drought should therefore have only minor effect on growth between sites. However, we will address this more clearly also in the discussion in the revised version.

467 – Transplant experiments are also a good approach for this and should not be ignored (e.g. Tito et al 2020) but may be limited to one or few species.

There is also a lot of work into temperature sensitivity measured as leaf temperature tolerance and safety margin. I would change a bit the phrase to be more specific to what you want to say.

**Response:** We state here that current knowledge is "primarily" based on observational studies controlled warming experiments. Transplant experiments is essentially the same as the Rwanda TREE approach (but at smaller spatial scale), and since these studies are very few for tropical trees they do make up a third body of evidence and thus do not qualify into this sentence. With respect to leaf level temperature studies, these are either part of observational or controlled experimental studies and thus do not conflict with the current wording. To justify the choice of references we will reword to "…*temperature sensitivity of tropical tree growth*…" at the beginning of the sentence.

477 – pattern was or patterns were

**Response:** Thanks for spotting this, it should be "patterns were".

482 – Moreover, other studies in the photosynthesis of Rwandan forests showed…

**Response:** Thanks, the text will be changed accordingly.

531- the mortality driven thermophilization is a mechanism at the community level. This study calculated the Community thermal index using the species optimum temperatures and their relative basal areas, and then its change through time. A change in CTI is easier/more obvious by the mortality of large trees than by the addition of new ones given their effect of the relative basal area. So I would not necessarily think that it is a big contradiction to your results of mortality per se, they are different approaches.

**Response:** Both growth and mortality contribute to the basal area-weighted index used in this study, but there is no "addition of new ones" since we planted seedings. It is therefore true that our approach differs from previous studies (Fadrique et al., Duque et al.), but still lack of statistical difference between the two origin elevation groups is contrasting with their findings.

537- Amazonian plots with tropical forest? Not sure why the explanation, is this for the Australian plots?

**Response:** This sentence refers to two different studies, one on plots in Australian tropical forest and another in plots of Amazonian forest. Note that the first study mentioned in the Amazonian forest (line 534-536) is different from the study mentioned on line 537.

Fig S1, S2 ,S3- I think you should mention here that the irrigation/nutrient experiment only started in 2019-9 and it had no effect on your results -  to remind that it is not a variable taken into account in the manuscript

**Response:** Yes, we will give this information also in the legends of Fig S1-S3.

---

## Author Comment (AC2)

**Comments from referee 2** (in blue)

General comments

This is a very interesting and timely paper on an important subject. The authors make use of the exciting Rwanda TREE experiment to assess the performance of a number of tree species from different climates of origin and different successional stages (early and late successional). These data are very novel, covering a large number of species and able to assess closely temperature effects with limited influence of drought. The findings are logical and supported by other literature, showing a higher sensitivity of late successional species and high elevation species to temperature, as measured with data on growth, mortality and contribution to basal area. There are some issues around the interpretation due to the experimental rather than observational nature of the study, but these are mostly sufficiently discussed and acknowledged by the authors.

**Response:** Thank you very much for appreciating our study. Thanks also for all constructive suggestions for improvements.

Specific comments

The main issue I found with the paper is that the site is planted with early and late successional species, and therefore in a high light environment it is possible there would be poor performance of the late successional species. However, this is raised and discussed in the paper, and the crucial thing is that this would be the case at all the sites, and therefore by looking at differences across the sites at different temperatures, this becomes less important. While this is in the discussion, I think it might be useful to add something on this, justifying the approach, in the Introduction so the reader is not wondering about it throughout.

**Response:** Thanks for this suggestion. We will add a sentence on this in section 2.3 about Plant material.

I also think that because these are experimentally planted sites, it is harder to consider that composition is really being assessed. I think the study brings great insight into how different species respond, and how this affects composition within the sites, and the knowledge gained can be generalised to natural systems, but I think there needs to be a little more conservative language, for example line 134 could be phrased "tree community composition within the experimental sites".

**Response:** Thanks for the comment. We will change the wording accordingly.

There is a lot of detail in the analysis, perhaps looking at too many variables. I am not sure the relative growth rate results are needed in addition to the analysis of final diameter and height. Unless the authors think this adds necessary insight, I suggest to remove it, or include in supplementary material. Some technical clarifications on the analysis are needed (see below).

**Response:** As the initial seedling sizes of different species were slightly different, we also included relative growth rates to show that the observed interspecific variations in height and diameter also to a large extent was independent of variation in initial sizes between species. We think the relative growth rate information therefore remove potential doubts about the validity of the results. We therefore suggest keeping this information as it is.

Technical comments (typos and small clarifications)

line 30. not much -> had little effect

**Response:** We prefer to keep the wording as it is.

line 64. at -> within

**Response:** This will be changed accordingly

line 87. is -> are

**Response:** This will be changed accordingly

line 129. What is the difference between Afromontane and African highland? Please define.

**Response:** We agree that the differences is not clear. We will therefore change it in this sentences to only include "African highland" to emphasise that the selected species are found at high elevations.

line 150. with -> by, check spelling on approximately

**Response:** This will be changed accordingly

line 175-180. This section could be written more clearly/simply. Please state what water and nutrient treatments were applied in chronological order. It seems the plants were watered except for one dry season – has this one dry season had an impact on results?

**Response:** We will change this paragraph as follows based on your recommendations: "*The 18 plots allowed for a full factorial experimental design, with three water levels and two fertility levels and a replication of three plots for each of the six treatment combinations. Before mid-July 2019, all trees at all sites received water when needed, irrespective of the subsequent planned water treatment. During the first dry period in July-August 2018 all trees were manually irrigated, while all plants were exposed to the dry period from mid-July to end of August 2019. The water and nutrient treatments started in September and November 2019, respectively. No significant (P > 0.05) effect on diameter and height growth of the treatments was observed during this period likely because September to December is within the rainy season and we therefore use averages of all plots. Site maps with experimental design are presented in Figures S1-S3.*"

Regarding the possible effect of the 2019 dry season this has not been analyses in this study as the there is a dry period at all sites. However, there are indications in the data shown in Figure S5 and S6 that the increment in both D and h are lower between census 6 and 7 (interval 6 - dry period 2019) compared to the before and after, but this is likely not influencing the overall result of the study. Furthermore, the mortality was not particularly high after the 2019 dry period (Table S6) indicating small effects.

No significant effect of the treatments on what?

**Response:** On tree diameter and height. We will add that.

line 201. Define wildling

**Response:** Wildling is naturally generated seedlings. The species that was noted as saplings in Table S1 were wildings, and we will change this in the table to be consistent between text and supplement information.

section 2.5 This section could be written more concisely.

**Response:** We will try to make it a bit more concise

line 248. know -> known

**Response:** Thanks, this will be changed accordingly

line 259. 3+ superscript

**Response:** Thanks, this will be changed accordingly

line 290. The basal area approach is somewhat confusing, but I understand the intention to standardize for different initial number of stems of each species group, but I think it could be explained better. The text mentions the dead tree contribution to basal area – presumably here considering the loss of trees to basal area composition, rather than basal area of dead trees? Clarify that only living trees are used for basal area. The approach to considering mortality impact on basal area was not easy to understand.

**Response:** Thanks for point this out. We will make it clear that only living trees were included in the basal area.

line 301. Here and other places it looks like there is a typo with repeated D-RGR_D10-25.

**Response:** No, it is not a typo. The D-RGR means that relative growth rates were calculated from diameter increments, and D10-25 means that it was bas on trees in size classes of 10-25 mm diameter. We will consider to remove the second D.

line 301. It needs to be specified that the D and h information is used only from the final census, so looking at the total effect of the two year study period. For the growth rates, is this averaged across all the census intervals?

**Response:** Thanks, you are right. To make this clear we will add that D and h were analysed only from the 8 census and number of stems from the 7 census, while all RGR values were analysed as average values of all intervals between all 8 censuses.

line 302. Here mentions both species and species group for the ANOVA models. Presumably this is then two separate analyses? The results seem to just report statistical results of species (e.g table 3, figure 2) and not species groups. If they are not included, perhaps remove mention on species groups here. If the are included, differences between groups could be added to Figure 2 with another set of letters.

**Response:** Thanks, groups should be deleted from this sentences and should read. "*…with site and species/species group as fixed factors.*"

line 307. Using plot level means. This is ok, but individual tree data could be used within mixed effects models, with a random effect for block. This would account for multiple trees within the same block.

**Response:** Yes, true but it would also add another source of variance and potentially lead to lower level of homogeneity in the dataset.

Table 3. Details of the anova results here and in other tables could be moved to the Supplement.

**Response:** Yes, if the editor suggests to do that we will move the details to the supplement. However, some journals prefer to show the details also in the main text.

Fig 3. The axis labels are difficult to read.

**Response:** We will increase the font size of the labels.

line 369/370. Are these numbers referring to D and h or ME and LE? Not clear.

**Response:** To make this clear we will slightly change this sentence: *On average, Dbase and h after two years for ES species* *increased by* *12% and 43% at the ME and LE sites, respectively, compared to the HE site.*

line 378. species name spelling and formatting needs correction.

**Response:** We don't understand what's wrong with spelling and formatting.

Fig4. X-axis could be relabelled with month/year.

**Response:** We will add year/month to the top x-axis and measuring intervals on the bottom x-axis to make this clear.

Line 415. Why is this important, does it mean it is before the irrigation was stopped and therefore that the results are temperature rather than moisture driven?

**Response:** We don't understand this comment in relation to the text on line 415. Has the line number been wrong in the version sent to this reviewer?

line 481. support -> supports

**Response:** Thanks, this will be changed accordingly

line 505. There is interesting discussion here. Over time, with a more closed canopy the LS species could increase following successional process, is this relevant here?

**Response:** This is likely but here we rather want to make the point that if large LS species trees are disfavoured at early successional stage, then that will have lasting effects into later successional stages.

line 586. divers -> diverse

**Response:** Thanks, this will be changed accordingly

line 593. The text could be more explicit about the implications of the results for biodiversity, carbon, species selection, e.g. which species types does this research suggest would be best for restoration projects.

**Response:** The Conclusions section is perhaps not the best place to develop this but we will try to do it elsewhere, if there are such places. The importance of large and long-lived LS trees for carbon storage is obvious, and LS trees with larger seeds/fruits are also important for many species. For plantation species selection it is more difficult to say as there are many different purposes for this.

---

## Author Response (AR1)

Dear Editor,

Below we have responded to all comments made by you and the referees as well as how we have revised the manuscript following these responses. All responses below are marked in blue and **line number in this response refer to the document with track changes.** All changes in the text are visible as "track changes".

Best wishes

Göran Wallin

Public justification (visible to the public if the article is accepted and published):

Dear authors,

Thanks for addressing all the comments made by the reviewers. Following I include some minor points that I hope they will help to improve the manuscript:

In the abstract, please add a sentence about the use of an elevation gradient as proxy of warming.

**Response:** This has been changed accordingly. Line 25.

Include in the introduction a sentence about the use of an elevation gradient as warming experiment.

**Response:** We believe that it already is a sentence with this meaning (line 147-148): "The elevation gradient was used as a proxy for possible future warming and irrigation was applied to compensate for variation in rainfall along the gradient"

The distinction between species groups is somehow confusing. There are two species functional groups (ES and LS) from two different regions (TMF and LVTF). From the research questions it is not obvious what the expectation for each group is. For instance, would LVTF species outperform TMF ones? Apart from this, it would have been interesting to categorize species into those that fall within their distribution range or not, if that is applicable.

**Response:** We think that we already have formulated expectations for different categories in the hypotheses. In hypothesis 1 for early and late successional species and in hypothesis 2 for montane and transitional forest. We have rephrased H#2 to make the classification of the two elevation groups clearer. Regarding the last suggestion, we think a broader range of species elevation origins and sites need to be included to make such analysis interesting. Line 155-157.

In Fig. 1, it seems that there is something strange with the Rubona site. To me it looks like a better control than Sigira.

**Response:** The likely reason that it looks a bit strange in Fig. 1 is that the axes for upper and lower elevation ranges don't cover the elevations of all sites. We have now increased the scale to include all sites on both axes and hope that this will Figure 1 now is better connected to the experiment. clearer. Regarding control choice of control, we have motivated this more in detail at the end of section 2.1 of the revised version.

As suggested by reviewer #2, elaborate con the community composition analysis. Explain how BAabs and BAfrac were calculated according to Eq.3.

**Response:** To make this clearer, we have added that only living individuals were included in the BA calculation and that the BAabs and BAfrac of the four species groups represented approximately 25% each, at the beginning of the experiment. Line 322 and 332-333.

In addition, the term community composition is a bit misleading because it suggests a "natural" community and that biodiversity analysis were undertaken. However, the study is based on a plantation experiment, which is not a proper community. I would suggest renaming this section to something like 'species groups distribution'. In addition, although the results were corrected by the initial values it is not easy to interpret them. Comparing starting and ending values will give more insight.

**Response:** We believe that also plantations have a "community composition", not only natural forests, and **_would prefer_** to use community composition. In case you think this is wrong, we suggest to change it to tree stand composition as we have done in the revised version.

In the statistical analysis, using an ANOVA to analyse number of stems and mortality is not the best option as they are counts and censored data, respectively. A GLM with a Poisson distribution and a survival analysis can be better options.

**Response:** Thanks for this suggestion. We agree that a Poisson regression is a better method for statistical analysis of count data as for mortality. We have now implemented this method on the mortality data. The results are however, mostly the same so no re-interpretation of data was necessary. We have not included a survival analysis as we have not aimed in this paper to in detail analyse the change in mortality with time. Line 350-353.

In section 490-505, another possibility of the lower performance of LS species could be that the experimental set-up does not trully mimic the natural conditions for this species. Apart from light, high densities are more common for ES species. However, LS species usually grow better at low densities because the other species has been filtered by light or other factors.

**Response:** Yes, it is true that stand density may affect early and late successional species differently. However, we believe this has not affected the composition of the species in the current study as plantations were still relatively open. However, this is something to consider in future studies. Thanks for this idea.

Finally, the special issue explores the potential of coupling experiments with models. I would encourage the authors to succinctly explain the potential of their experiments in a modelling approach to tackle the issues indicated in the last comment of reviewer #2, in particular the one related to ecosystem function and the response to global change components.

**Response:** As a response to this, we added a short discussion of potential implications at the end of the conclusion section, and also change the title of the sections to conclusions and implications. Line 654-669.

Best,

Víctor Rolo

**Comments from referee 1**

In this manuscript the authors describe the results of a large manipulative experiment where multiple tree species were planted at three different elevations in order to observe the effect of temperature change in their growth and mortality. In addition, these species represented different successional strategies (Early and Late) and different forests of origin (montane vs transitional). I find the experiment very impressive. The manuscript is very well-written and presented so I want to congratulate the authors. I do however, think that the story is quite complicated and the constant use of acronyms does not help to simplify it. The intro needs a few adjustments to make it shorter and more concise (see below) but it is on the Methods/Results and Discussion where it gets harder to follow and stay engaged. These are a few suggestions to make the manuscript more engaging:

o        Try to reduce the number of acronyms or modify the current ones for something more explicit, for example: LVTF- Transitional forests, TMF – Montane forests; ES- Early-S, LS- Late-S

o        Include a diagram that explains the experimental set up with the three elevations, species origins, data recorded (could be in Supplement)

o        Include a summary figure with your main results. The figures (for example fig.2 and 3) have a lot of information and it is hard to focus on what is important and what does it mean.

Something that I do not understand is the role of the higher elevation site as the control site. That is mentioned in the methods but I do not see much discussion around this fact.

The title undersells the study. That could be the title for an observational study. I think it should reflect the enormous experimental work and novelty of it.

**Response:** Thank you very much for appreciating our study. Thanks also for all constructive suggestions for improvements.

We have now in most case of the text used transitional forest and montane forest to replace the acronyms LVTF and TMF and we have spelled out early and late successional species in most case in the revised manuscript. The acronyms are still used in figures and tables with full explanation in legends, but also in a few cases, mostly when it is referred to groups combining successional stages and origin of species.

We will consider to add a diagram explaining the experimental set-up for the supplement or as a graphical abstract, but have not done that yet.

We agree that Fig. 2 and 3 have a lot of information, but we have tried to synthesis this information in Fig. 5, 6 and 7 so we believe that additional figures are not needed, so no action on this.

The reference to the highest site as the control site is accurate for the species originating from montane rainforest; for these species the other sites represent a warming scenario. For the species originating from transitional rain forest it is not that simple. The revised text now reads: *"Although many of the selected species are distributed in both montane and transitional forest, the HE site (Sigira) is considered as the control site in this experiment since today's remaining natural forests are predominantly montane and all species except one can be found at >2000 m elevation (Figure 1). Furthermore 18 out of 20 species used in this experiment naturally grow in the neighbouring NNP, ranging from 2950 down to 1600 m a.s.l. (Fisher and Killman, 2008; Nyirambangutse et al., 2017). With the HE site as control, the ME (Rubona) and LE (Makera) sites represent two different warming scenarios."* See line 177-183.

Thanks for suggesting a stronger title. As a revised title we suggest: Thermophilization of Afromontane forest stands demonstrated in an elevation gradient experiment.

Some line comments:

Abstract

Very well written. The only doubt I have is at what stage where the plants transplanted (seedling, sapling...)

**Response:** It was seedlings (except for two species for which cuttings were used but in a "seedling size"). This has been added in the abstract.

Introduction

45- negative effects on xxxx and where?

**Response:** Thanks for the comments, we have added this information: "…*warm and dry conditions during El Niño years have caused negative effects on tree growth in Central America, Amazonia, West and Central Africa and South-West Asia as well as increased tree mortality in Amazonia (Clark et al., 2003; Lewis et al., 2011; Rifai et al., 2018).*" Line 48-50.

50 - in the field? Need to be more specific, In mountains heat and drought do not always covary

**Response:** We have removed "*in the field*" as it may confuse, and more clearly point out that heat and drought not always covary by changing this sentence: "*However, due in part to common co-variation of heat and drought, the direct effect of warming on tropical forests remains unclear*". Line 54.

59 – Large variability… This sentence is unclear needs to be reorganized.

**Response:** We have revised, and these sentences now reads: "*Large variability across studies and species has been observed, from positive to negative effects (e.g. Slot & Winter, 2018; Dusenge et al., 2021; Wittemann et al., 2022). This may reflect large variation in species origin temperature zone and how this compares to experimentally applied temperature treatments, with positive effects being more likely if origin climate is rather cool and the warming treatment is modest.*" Line 63-67.

64 – You may want to read Tovar et al 2022

**Response:** Thank you for bringing this recent publication to our attention. We have cited Tovar et al 2022 to support the statement on earlier line 64, now line 70.

65 – The lower elevation limit of TMF varies widely between and within continents so I am not sure what this >1000 m a.s.l. refers to

**Response:** We agree that the TMF elevation limit is highly variable. However, most estimates of their area globally are based on a general lower elevation limit. We think it is important to give the reader an idea of the extension of such forests so we used and relatively recent estimate from Spracklen & Righelato, 2014 who used 1000 m a.s.l. as a limit. To clarify this the text has been modified as follows: "*TMFs occur at all continents within the tropical biome. The lower elevation limit varies*

*widely between and within continents but using a general lower limit of 1000 m a.s.l., the cover has been estimated to 8% of the total tropical forest area (Spracklen & Righelato, 2014)."* Line 71-72.

70 – why species on the higher elevations are further away from their thermal optimum? You imply that they could tolerate a higher increase in temperature than lower elevation plants, but you need to justify that. For example, Leon-Garcia and Lasso 2019. Although here they go all the way to the paramo ecosystem.

**Response:** We agree that this statement should be justified by a citation. Thanks for the suggested reference, but we think that Feeley et al 2020 in *Frontiers in Forests and Global Change,* better support this statement so we have added that reference instead. Line 78-79.

83-90 The intro is great until this paragraph. I think it is a bit repetitive, and there is one minor point that could be made more clearly and briefly. Maybe as a first line of the next paragraph a sentence linking the functional strategy with different chances to survive and then growth forms?

**Response:** We agree that this paragraph is a bit redundant and has removed it completely. Line 91-98.

o        I am more familiar with using ES and LS to refer to species but primary or old-growth and secondary forests to the forests. I think better to keep them separate.

**Response:** We agree that primary and secondary forests are more commonly used and we have changed to this terminology in the revised version. Line 100.

o        I don't like the sentence: remains uncertain. is it the forests? The species?.

**Response:** We agree that this is unclear and have changed this sentence to: "*However, it remains uncertain if the early successional species are winners or loser in a future warming climate and thus if climate change will amplify the expansion of secondary forests will be amplified or not".* Line 105-106.

o        Also, these experimental indications – how are they different from what you are doing here? Are they greenhouse experiments, models? Better to indicate so you can highlight the novelty of your paper.

**Response:** Thanks for this suggestion. We have added a sentence that these studies either only included few species representing ES and LS strategies, use artificial heating in chambers or infra-red heaters or only have looked at physiology. Line 109-110.

100 – defining ES species here is a bit late, should come at the beginning of the paragraph.

**Response:** We have moved the text about fast and slow growing species to the beginning where ES and LS species are defined as acquisitive vs. conservative species. Line 100-104.

100 – suspectable= susceptible? Susceptible

**Response:** Thank you for spotting this typo. It has been changed to susceptible.

Overall, this paragraph also needs some reorganizing and trimming

**Response:** We think that the paragraph is mostly good, but as said above we have moved the information about slow and fast growing species to the first part and also made a few other revisions in the text to improve the paragraph. Line 100-116.

105-115. This seems like a good point but is it important enough to be a whole paragraph? I think the intro is really long and this is something that you should cut.

**Response:** Indeed, we think it is an important point and based on comments from referee 2 and the editor asking for us to elaborate more on implications, we kept this paragraph in the introduction, but reduced the length slightly, to be able to refer back to these issues at the end of the paper. Line 117-130.

116 – I feel now we are back on track. This brings the story back to line 81 – I would probably talk about the growth strategies after this paragraph or even after the next one where you explain your objective. The reader needs to know early on the intro what you are doing on the manuscript, and it is not clear until here.

**Response:** Thanks for the suggestion, however, we think that the growth strategies are well placed in the introduction, especially after removing the paragraph before the growth strategies as it now better connects to the previous paragraph.

125 – identical plant material- not sure what it means

**Response:** We will change this to "*genetically similar plant material*"**.** Line 140.

129 – at what stage are the trees transplanted? Saplings?

**Response:** The plants were in seedling sizes (propagated from seeds, or collected as naturally generated seedlings in the forest and one species propagated as cuttings) when transplanted from a central nursery (at the mid elevation) to the sites but most species developed quickly into sapling sizes, while a few stayed in a seedling size for quite long time. We have now added that "*Seedling sized trees were planted…*" in the paragraph before the hypotheses. Line 146.

135 – are lower elevation species planted on even lower elevations? I get that from the hypothesis. Maybe include the elevational range of the species in the objective?

**Response**: The lowest site is below the "transitional rainforest" category so we would say YES, there is a warming treatment for all species. For some species, the distribution range covers the lowest site, but dominating distribution is from the "transitional rainforest" area. We have specified the elevation origin of both species from the montane and the transitional forests at the end of the introduction of the revised version.

Overall, I would reduce the background information on precipitation and drought because now I realize that you are irrigating to isolate the effect of temperature, so better to focus on that.

**Response:** We believe that it is important to give background information about both drought and temperature as they are closely connected. In our case the soil drought effect is small (expect the 2019 drought period) while still the there is an VPD effect.

Methods

150 – increased/decreased by

**Response:** This has been changed accordingly

159 – I don't understand why that site is the control at 2400 because the species grow nearby at (potentially) 1600m which is as low as the mid-elevation site.

**Response:** As pointed out earlier, we have rephrased the statement about the control site, see the end of section 2.1.

In Table 1 PNV of HE should be the full name to make it easier to link with the text (TMF)

**Response:** Changed accordingly

224 – I do wonder about the effect of solar radiation. The HE site is thus the control but also the more shaded one.

**Response:** Yes, solar radiation is slightly lower at the HE site, This is clearly mention in material and methods and possible effects of radiation is discussed on line 543-549 in the revised version. We also added a sentence on this in material and methods as a response to referee 2, see Line 265-267.

It is overall a bit confusing the species origin vs the planting sites. And where is the control evaluated?

**Response:** As indicated above, the statement about control site has been revised. The sites are compared internally rather than using one control site in the statistical analysis.

Results (see comments above)

Discussion

405 – This drought period, was it a problem with the irrigation system? Has it been mentioned before? can it have an effect on other variables?

**Response:** On line 201-203 it is mentioned: "while all plants were exposed to the dry period from mid-July to end of August 2019." The drought in 2019 was therefore intentional and not linked to a problem with the irrigation system. As mentioned on line 250-252, the relative seasonal distribution of precipitation was similar at all sites, and the 2019 drought should therefore have only minor effect on growth between sites. In the revised version we have rephrased both in the material and methods (line 200-209) as well as in the discussion to make this clearer (line 453-454).

467 – Transplant experiments are also a good approach for this and should not be ignored (e.g. Tito et al 2020) but may be limited to one or few species.

There is also a lot of work into temperature sensitivity measured as leaf temperature tolerance and safety margin. I would change a bit the phrase to be more specific to what you want to say.

**Response:** We state here that current knowledge is "primarily" based on observational studies controlled warming experiments. Transplant experiments is essentially the same as the Rwanda TREE approach (but at smaller spatial scale), and since these studies are very few for tropical trees they do make up a third body of evidence and thus do not qualify into this sentence. With respect to leaf level temperature studies, these are either part of observational or controlled experimental studies and thus do not conflict with the current wording. To justify the choice of references we will reword to "…*temperature sensitivity of tropical tree growth*…" at the beginning of the sentence. Line 515

477 – pattern was or patterns were

**Response:** Thanks for spotting this, it has been revised to "patterns were".

482 – Moreover, other studies in the photosynthesis of Rwandan forests showed…

**Response:** Thanks, the text has been changed accordingly.

531- the mortality driven thermophilization is a mechanism at the community level. This study calculated the Community thermal index using the species optimum temperatures and their relative basal areas, and then its change through time. A change in CTI is easier/more obvious by the mortality of large trees than by the addition of new ones given their effect of the relative basal area. So I would not necessarily think that it is a big contradiction to your results of mortality per se, they are different approaches.

**Response:** Both growth and mortality contribute to the basal area-weighted index used in this study, but there is no "addition of new ones" since we planted seedings. It is therefore true that our approach differs from previous studies (Fadrique et al., Duque et al.), but still lack of statistical difference between the two origin elevation groups is contrasting with their findings.

537- Amazonian plots with tropical forest? Not sure why the explanation, is this for the Australian plots?

**Response:** This sentence refers to two different studies, one on plots in Australian tropical forest and another in plots of Amazonian forest. Note that the first study mentioned in the Amazonian forest (line 589-591) is different from the study mentioned on line 592.

Fig S1, S2 ,S3- I think you should mention here that the irrigation/nutrient experiment only started in 2019-9 and it had no effect on your results - to remind that it is not a variable taken into account in the manuscript

**Response:** Thanks for this suggestion. We have added this this information also in the legends of Fig S1-S3.

**Comments from referee 2**

General comments

This is a very interesting and timely paper on an important subject. The authors make use of the exciting Rwanda TREE experiment to assess the performance of a number of tree species from different climates of origin and different successional stages (early and late successional). These data are very novel, covering a large number of species and able to assess closely temperature effects with limited influence of drought. The findings are logical and supported by other literature, showing a higher sensitivity of late successional species and high elevation species to temperature, as measured with data on growth, mortality and contribution to basal area. There are some issues around the interpretation due to the experimental rather than observational nature of the study, but these are mostly sufficiently discussed and acknowledged by the authors.

**Response:** Thank you very much for appreciating our study. Thanks also for all constructive suggestions for improvements.

Specific comments

The main issue I found with the paper is that the site is planted with early and late successional species, and therefore in a high light environment it is possible there would be poor performance of the late successional species. However, this is raised and discussed in the paper, and the crucial thing is that this would be the case at all the sites, and therefore by looking at differences across the sites at different temperatures, this becomes less important. While this is in the discussion, I think it

might be useful to add something on this, justifying the approach, in the Introduction so the reader is not wondering about it throughout.

**Response:** Thanks for this suggestion. We have added the following sentence in the materials and methods (section 2.4) *"Differences in light climate may influence species with early and late successional strategies differently (Ntawuhiganayo et al 2020), but we believe that the on average 8.5% lower radiation at HE site compared to LE site will have minor effects."* Line 265-267.

I also think that because these are experimentally planted sites, it is harder to consider that composition is really being assessed. I think the study brings great insight into how different species respond, and how this affects composition within the sites, and the knowledge gained can be generalised to natural systems, but I think there needs to be a little more conservative language, for example line 134 could be phrased "tree community composition within the experimental sites".

**Response:** Thanks for the comment. We have changed the wording accordingly in the revised version. Line 151.

There is a lot of detail in the analysis, perhaps looking at too many variables. I am not sure the relative growth rate results are needed in addition to the analysis of final diameter and height. Unless the authors think this adds necessary insight, I suggest to remove it, or include in supplementary material. Some technical clarifications on the analysis are needed (see below).

**Response:** As the initial seedling sizes of different species were slightly different, we also included relative growth rates to show that the observed interspecific variations in height and diameter also to a large extent was independent of variation in initial sizes between species. The RGR data is also averaging over a time period in which the size criteria is met, while the *D* and *h* data only consider the final values. We think the relative growth rate information therefore remove potential doubts about the validity of the results. We have therefore kept this information as it is.

Technical comments (typos and small clarifications)

line 30. not much -> had little effect

**Response:** We prefer to keep the wording as it is.

line 64. at -> within

**Response:** This text has been rephrased according to Referee 1 comments and this comment is therefore not valid any longer.

line 87. is -> are

**Response:** The paragraph has been deleted.

line 129. What is the difference between Afromontane and African highland? Please define.

**Response:** We agree that the differences is not clear. We have therefore changed this to only "African upland" to emphasise that the selected species are found at high elevations. We think that upland is a bit more appropriate term than highland as it is not necessarily mountainous areas. Line 145.

line 150. with -> by, check spelling on approximately

**Response:** This has been changed accordingly.

line 175-180. This section could be written more clearly/simply. Please state what water and nutrient treatments were applied in chronological order. It seems the plants were watered except for one dry season – has this one dry season had an impact on results?

**Response:** We have changed this paragraph as follows based on your recommendations: "*The 18 plots allowed for a full factorial experimental design, with three water levels and two fertility levels and a replication of three plots for each of the six treatment combinations. Before mid-July 2019, all trees at all sites received water when needed, irrespective of the subsequent planned water treatment. During the first dry period in July-August 2018 all trees were manually irrigated, while all plants were exposed to the dry period from mid-July to end of August 2019. The water and nutrient treatments started in September and November 2019, respectively. No significant (P > 0.05) effect on diameter and height growth of the treatments was observed during this period likely because September to December is within the rainy season and we therefore use averages of all plots. Site maps with experimental design are presented in Figures S1-S3.*" Line 199-209.

Regarding the possible effect of the 2019 dry season this has not been analyses in this study as the there is a dry period at all sites. However, there are indications in the data shown in Figure S5 and S6 that the increment in both D and h are lower between census 6 and 7 (interval 6 - dry period 2019) compared to the before and after, but this is likely not influencing the overall result of the study. Furthermore, the mortality was not particularly high after the 2019 dry period (Table S6) indicating small effects.

No significant effect of the treatments on what?

**Response:** On tree diameter and height. This has been added.

line 201. Define wildling

**Response:** Wildling is naturally generated seedlings. The species that was noted as saplings in Table S1 and wildings in the text. We will change to "*naturally generated seedlings*" at both places. Line 230.

section 2.5 This section could be written more concisely.

**Response:** We have deleted a few sentences containing information that might not be necessary to include.

line 248. know -> known

**Response:** Thanks, this has been changed accordingly

line 259. 3+ superscript

**Response:** Thanks, this has been changed accordingly

line 290. The basal area approach is somewhat confusing, but I understand the intention to standardize for different initial number of stems of each species group, but I think it could be explained better. The text mentions the dead tree contribution to basal area – presumably here considering the loss of trees to basal area composition, rather than basal area of dead trees? Clarify that only living trees are used for basal area. The approach to considering mortality impact on basal area was not easy to understand.

**Response:** Thanks for point this out. We have added that only *"living individuals"* (Line 322) were included in the basal area and that the BAabs and BAfrac of the four species groups represented approximately 25% each, at the beginning of the experiment. (Line 332-333).

line 301. Here and other places it looks like there is a typo with repeated D-RGR_D10-25.

**Response:** No, it is not a typo. The D-RGR means that relative growth rates were calculated from diameter increments, and D10-25 means that it was based on trees in size classes of 10-25 mm diameter. This has not been changed.

line 301. It needs to be specified that the D and h information is used only from the final census, so looking at the total effect of the two year study period. For the growth rates, is this averaged across all the census intervals?

**Response:** Thanks, you are right. To make this clear we have added that D and h were analysed only from the 8 census and number of stems from the 7 census, while all RGR values were analysed as average values of all intervals between all 8 censuses. Line 335.

line 302. Here mentions both species and species group for the ANOVA models. Presumably this is then two separate analyses? The results seem to just report statistical results of species (e.g table 3, figure 2) and not species groups. If they are not included, perhaps remove mention on species groups here. If the are included, differences between groups could be added to Figure 2 with another set of letters.

**Response:** Thanks, groups has been deleted from this sentences and now reads: "…*with site and species/species group as fixed factors.*" Line 336.

line 307. Using plot level means. This is ok, but individual tree data could be used within mixed effects models, with a random effect for block. This would account for multiple trees within the same block.

**Response:** Yes, true but it would also add another source of variance and potentially lead to lower level of homogeneity in the dataset.

Table 3. Details of the anova results here and in other tables could be moved to the Supplement.

**Response:** Yes, if the editor suggests to do that we will move the details to the supplement. However, some journals prefer to show the details also in the main text.

Fig 3. The axis labels are difficult to read.

**Response:** The font size of the labels has been increased.

line 369/370. Are these numbers referring to D and h or ME and LE? Not clear.

**Response:** To make this clear we have changed this sentence to: *On average, Dbase and h after two years for ES species underlined{increased by} 12% and 43% at the ME and LE sites, respectively, compared to the HE site.* Line 414-417.

line 378. species name spelling and formatting needs correction.

**Response:** We don't understand what's wrong with spelling and formatting.

Fig4. X-axis could be relabelled with month/year.

**Response:** We have added Year and month to the top x-axis and measurement interval on the bottom x-axis.

Line 415. Why is this important, does it mean it is before the irrigation was stopped and therefore that the results are temperature rather than moisture driven?

**Response:** We don't understand this comment in relation to the text on line 415. Has the line number been wrong in the version sent to this reviewer?

line 481. support -> supports

**Response:** Thanks, this has been changed accordingly

line 505. There is interesting discussion here. Over time, with a more closed canopy the LS species could increase following successional process, is this relevant here?

**Response:** This is likely but here we rather want to make the point that if large LS species trees are disfavored at early successional stage, then that will have lasting effects into later successional stages.

line 586. divers -> diverse

**Response:** Thanks, this has been changed accordingly

line 593. The text could be more explicit about the implications of the results for biodiversity, carbon, species selection, e.g. which species types does this research suggest would be best for restoration projects.

**Response:** As indicated in the response to the editors comments, we added a short discussion of potential implications at the end of the conclusion section, and also change the title of the sections to conclusions and implications. Line 654-669.

---

## Author Response (AR2)

Dear Editor,

This file show technical editing of text, tables and figures done after the paper was accepted. All
changes are visible in track changes mode accept the conversion of Tables from pictures to Text.
In summary:
1) Affiliations have been updated
2) A figure to give an overview of the experiment has been added (Figure S1), as requested by
   reviewer 1. Consequently the numbers of the supporting Figures have changed.
3) Tables have been converted from picture to text
4) Figures have been refined in Sigma plot
5) A technical terms have been changed.

Thanks!

Göran Wallin

[revised manuscript text omitted]